# MASK-DPO: GENERALIZABLE FINE-GRAINED FACTUALITY ALIGNMENT OF LLMS

**Yuzhe Gu**[1,2]   **Wenwei Zhang**[2†]   **Chengqi Lyu**[2]   **Dahua Lin**[2,3]   **Kai Chen**[2†]
[1]Shanghai Jiao Tong University   [2]Shanghai AI Laboratory
[3]MMLab, The Chinese University of Hong Kong
`{guyuzhe,zhangwenwei,lvchengqi,chenkai}@pjlab.org.cn`

## ABSTRACT

Large language models (LLMs) exhibit hallucinations (*i.e.*, unfaithful or nonsensical information) when serving as AI assistants in various domains. Since hallucinations always come with truthful content in the LLM responses, previous factuality alignment methods that conduct response-level preference learning inevitably introduced noises during training. Therefore, this paper proposes a fine-grained factuality alignment method based on Direct Preference Optimization (DPO), called Mask-DPO. Incorporating sentence-level factuality as mask signals, Mask-DPO only learns from factually correct sentences in the preferred samples and prevents the penalty on factual contents in the not preferred samples, which resolves the ambiguity in the preference learning. Extensive experimental results demonstrate that Mask-DPO can significantly improve the factuality of LLMs responses to questions from both in-domain and out-of-domain datasets, although these questions and their corresponding topics are unseen during training. Only trained on the ANAH train set, the score of Llama3.1-8B-Instruct on the ANAH test set is improved from 49.19% to 77.53%, even surpassing the score of Llama3.1-70B-Instruct (53.44%), while its FactScore on the out-of-domain Biography dataset is also improved from 30.29% to 39.39%. We further study the generalization property of Mask-DPO using different training sample scaling strategies and find that scaling the number of topics in the dataset is more effective than the number of questions. We provide a hypothesis of what factual alignment is doing with LLMs, on the implication of this phenomenon, and conduct proof-of-concept experiments to verify it. We hope the method and the findings pave the way for future research on scaling factuality alignment. Code is available at `https://github.com/open-compass/ANAH`.

## 1 INTRODUCTION

Large Language Models (LLMs) have demonstrated impressive performance across various tasks (Kamalloo et al., 2023; Sun et al., 2023; Chen et al., 2024). Even though, LLMs still face a concerning issue: *hallucination*, in which they generate plausible-sounding yet inaccurate or nonsensical information (Ji et al., 2022; Bang et al., 2023) in response to user queries, particularly those demanding extensive knowledge. The hallucination phenomenon has two main characteristics (Ji et al., 2024; Mishra et al., 2024) that significantly impede the real-world application of LLMs: 1) the unfaithful or nonsensical information always exists with truthful contents in the LLM responses, posing a challenge to accurately and effectively detect and mitigate them; 2) LLMs exhibit hallucinations across various domains and tasks, but it is difficult and impractical to thoroughly mitigate them by exhausting all real-world knowledge. The prevalence of these two issues underscores the need to develop effective and generalizable methods for fine-grained factuality alignment to reduce hallucinations in Large Language Models (LLMs).

Existing methods (Tian et al., 2023; Lin et al., 2024; Zhang et al., 2024b; Chen & Li, 2024) mainly apply preference learning-based method, especially Direct Preference Optimization (DPO) (Rafailov et al., 2024), to align the internal knowledge of LLMs to the facts. These approaches utilize the

---

† Corresponding author

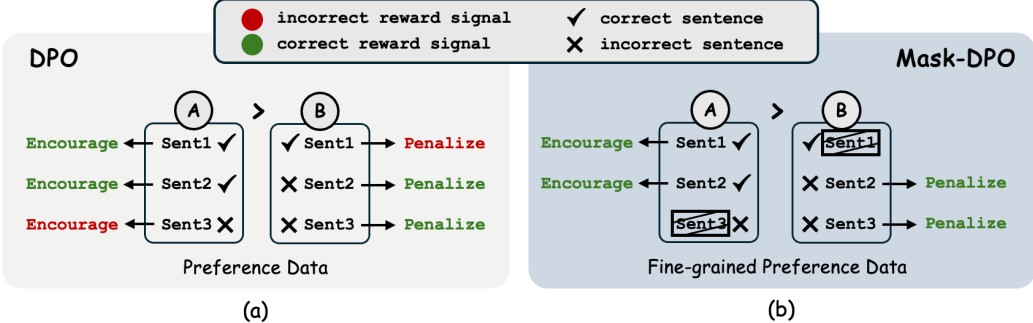

Figure 1: **Comparison between DPO and Mask-DPO.** Vanilla DPO (a) inadvertently encourages and penalizes all the content in the preferred and non-preferred samples, respectively, regardless of their correctness. Instead, Mask-DPO (b) incorporates sentence-level facticity into the mask signal, preventing incorrect reward signal, which resolves ambiguity in preference learning.

response-level factuality for constructing pairwise preference data for preference optimization, which aims to maximize the probability of preferred samples with higher factuality while minimizing the probability of non-preferred samples with lower factuality. However, since both factually correct and incorrect sentences are often mixed within a single response (Ji et al., 2024; Mishra et al., 2024), vanilla DPO inadvertently and inevitably encourages incorrect information in the preferred samples and penalizes truthful ones in the non-preferred samples (Figure 1(a)). Such ambiguity in the learning exists as long as the training samples are not completely wrong or correct, which is often the case, ultimately reducing the effectiveness of factuality alignment.

To address the issue, this paper proposes a fine-grained factuality alignment framework, named Mask-DPO. As shown in Figure 1, unlike vanilla DPO, Mask-DPO leverages sentence-level factuality information as a mask signal to resolve the ambiguity in preference learning. Specifically, in the preference data construction stage, Mask-DPO uses a fine-grained hallucination annotator (*e.g.*, ANAH-v2 (Gu et al., 2024)) to determine the factual correctness of each sentence, which will be used to guide the preference learning later. The responses that contain more and fewer factually correct sentences will be chosen as preferred and non-preferred samples, respectively, in a preference pair for learning. During the preference learning stage, the sentences that cause ambiguities in training, *i.e.*, incorrect sentences in the preferred samples and correct sentences in the not preferred samples, would be ignored, guided by the sentence-level mask annotated in the previous stage. Thus, Mask-DPO can avoid encouraging incorrect sentences when maximizing the probability of the preferred sample, and avoid penalizing correct sentences when minimizing the probability of the non-preferred sample.

Through extensive experiments, we demonstrate that Mask-DPO significantly enhances the factuality of LLMs and exhibits more effectiveness than DPO. Mask-DPO boosts the score of Llama3.1-8B-Instruct on the ANAH test set from 49.19% to 77.53%, surpassing the Llama3.1-70B-Instruct (53.44%) and DPO (68.44%), although the questions and topics in the test set are unseen during the alignment. Moreover, when testing the same model on the out-of-domain Biography dataset, Llama3.1-8B-Instruct aligned by Mask-DPO also exhibits excellent performance, whose FactScore is improved from 30.29% to 39.39%, reaching a level close to Llama3.1-70B-Instruct (40.47%).

We further study the scaling and generalization property of Mask-DPO, by scaling the training data through two dimensions: the number of different topics and the diversity of questions within the same range of topics. The experimental results show that scaling the number of topics is more effective for improving the factuality and generalization of models, in comparison to increasing the diversity of questions under the same topics, under the constraints of using the same amount of preference pairs. On the implication of this phenomenon, we hypothesize that each LLM learns a model-specific knowledge graph during training with the language modeling objective, where different topics and their corresponding information can be regarded as graph nodes, and the affinity between two nodes positively determines the probability of blurring their corresponding knowledge when the LLM responds to the questions related to either one of the topics.

Under this hypothesis, factual alignment on a topic essentially adjusts the affinity between the topic and its nearby nodes, which will also help to improve the factuality when the model answers questions

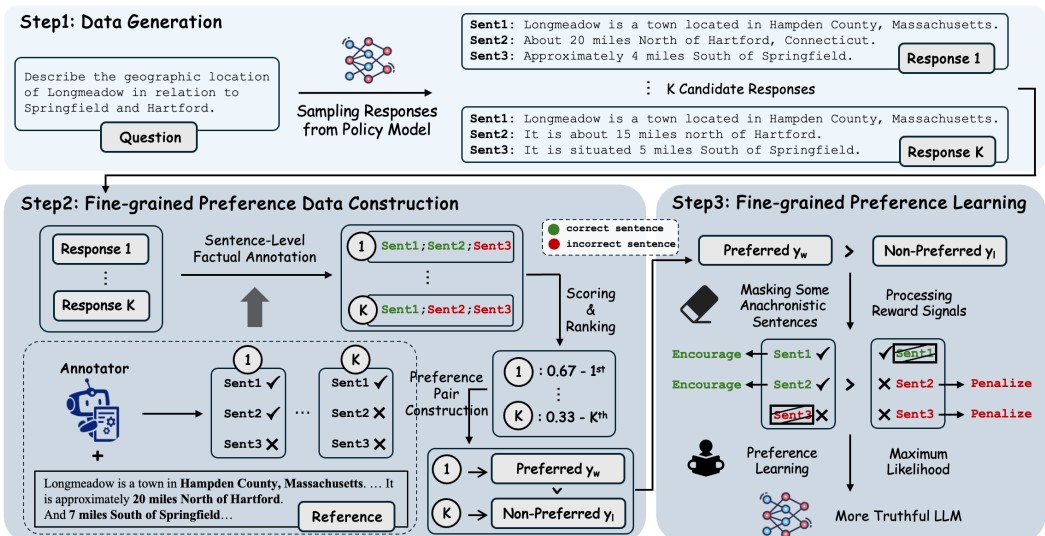

Figure 2: **The overview of Mask-DPO.** First, we sample K candidate responses for each question from the policy model. Then, we use a fine-grained hallucination annotator to perform a sentence-level factuality annotation on each response. We use the proportion of correct sentences out of the total number of sentences as the factuality score. We select the responses with the highest and lowest scores as preferred and non-preferred samples, respectively. Finally, we perform fine-grained factuality alignment on the policy model using such fine-grained preference data, where the reward signals to the sentences, *i.e.*, incorrect sentences in the preferred samples and correct sentences in the non-preferred samples, would be ignored.

of the nearby topics, even if they are not seen in the alignment. We conduct two proof-of-concept experiments to verify our hypothesis. The first one ablates different topic sampling strategies and shows that sampling the nearest topics to those in the test set based on the model-specific clustering can most effectively improve the factuality score on the test set. The comparison between the factuality of best-of-N responses of the model before and after factuality alignment further verifies that the internal knowledge of LLMs is indeed more aligned to the facts after factuality alignment, which has not been observed in previous experiments of preference learning on other domains such as reasoning (Havrilla et al., 2024). We wish Mask-DPO and our findings could pave the way for future research on scaling factuality alignment of LLMs.

## 2 MASK-DPO

Mask-DPO is built on a preference-based reinforcement learning framework (§ 2.1). While previous methods optimize preferences at the response level, our approach employs dense supervision at the sentence level and performs a fine-grained preference learning(§ 2.2), as shown in Fig. 2.

### 2.1 PRELIMINARIES

Reinforcement Learning from Human Feedback (RLHF) is a powerful alignment method for fine-tuning language models to enhance their robustness, factuality, and safety (Ouyang et al., 2022). In practice, given a prompt $x$ and a corresponding LLM response $y$, RLHF aims to maximize the following objective:

$$\max_{\pi_\theta} \mathbb{E}_{x \sim \mathcal{D}_p, y \sim \pi_\theta(\cdot|x)} \left[ r(x, y) - \beta \log \frac{\pi_\theta(y_l|x)}{\pi_{\text{ref}}(y_l|x)} \right], \tag{1}$$

where $\mathcal{D}_p$ represents the prompts dataset, $\pi_\theta$ is the policy model to be optimized, $\pi_{ref}$ is the reference model used for regularizing $\pi_\theta$ with Kullback–Leibler divergence and $\beta$ is a constant to control the degree of regularization. The reward model $r$ reflects human preferences, which takes a prompt and the corresponding response as input and outputs a scalar value.

However, training a reward model and integrating it into the overall pipeline can be quite complex (Zheng et al., 2023). To avoid this, Rafailov et al. (2024) proposed Direct Preference Optimization (DPO), which directly optimizes the policy model using preference pairs. Given a prompt $x$, a preference pair that contains $y_w$ and $y_l$ sampled from the policy model $\pi_\theta$, where $y_w$ has higher quality than $y_l$. DPO aims to maximize the probability of the preferred sample $y_w$ while minimizing the probability of the less desirable sample $y_l$. The optimization objective is formulated as:

$$\mathcal{L}_{\text{DPO}}(\pi_\theta; \pi_{\text{ref}}) = -\mathbb{E}_{(x,y_w,y_l)\sim\mathcal{D}} \left[ \log \sigma \left( \beta \log \frac{\pi_\theta(y_w|x)}{\pi_{\text{ref}}(y_w|x)} - \beta \log \frac{\pi_\theta(y_l|x)}{\pi_{\text{ref}}(y_l|x)} \right) \right] \tag{2}$$

where $\mathcal{D}$ is the pair-wise preference dataset and $\sigma$ is the sigmoid function.

Preference learning-based alignment has been proven effective for factuality tuning in mitigating the hallucination of LLMs (Tian et al., 2023; Lin et al., 2024; Zhang et al., 2024b). However, the optimization objective of Eq. 2 may not be entirely suitable for knowledge-based tasks. Taking the preferred sample $y_w$ as an example, we assume it has $N$ sentences, $i.e. y_w = \{s_i\}_{i=1}^N$. The log of its generative probabilities in Eq. 2 can be expanded as:

$$\log \frac{\pi_\theta(y_w|x)}{\pi_{\text{ref}}(y_w|x)} = \log \frac{\prod_{i=1}^N \pi_\theta(s_i|x, s_{1\sim i-1})}{\prod_{i=1}^N \pi_{\text{ref}}(s_i|x, s_{1\sim i-1})} = \sum_{i=1}^N \log \frac{\pi_\theta(s_i|x, s_{1\sim i-1})}{\pi_{\text{ref}}(s_i|x, s_{1\sim i-1})} \tag{3}$$

In such case, once there is a sentence $s_i$ in $y_w$ that contains incorrect facts, the generation probability of $s_i$ would be maximized when maximizing the generation probability of $y_w$. Similarly, the generation probability of the correct sentence in $y_l$ would be minimized when minimizing the $y_l$ generation probability. This affects the effectiveness of factuality alignment.

## 2.2 FINE-GRAINED FACTUALITY ALIGNMENT

**Fine-grained Preference Data Construction.** We perform fine-grained factuality annotation using a sentence-level hallucination annotator (or reward model). Given the prompt $x$ and its corresponding response $y$, which consist of $N$ sentences $s = \{s_i\}_{i=1}^N$, the reward model $\mathcal{A}$ annotate each sentence and give the fine-grained feedback $a = \{a_i\}_{i=1}^N$, which can be formulated as:

$$a = \{a_i\}_{i=1}^N = \{\mathcal{A}(s_i, x)\}_{i=1}^N = \mathcal{A}(y, x) \tag{4}$$

Here, we let $a_i = 0$ represent that sentence $s_i$ has no hallucination, and $a_i = 1$ represents that it has the hallucination.

Thanks to the fine-grained factuality annotation, we can now construct the preference dataset for alignment. First, for each prompt $x$, we perform top-k sampling to select $K$ candidate responses $y = \{y^i\}_{i=0}^K$ from the policy model. For each candidate, we annotate it according to Eq. 4 and obtain $\{x, y^i, s^i, a^i\}_{i=0}^K$. Next, we compute the factuality score for each candidate, which we define as the fraction of correct facts: $\frac{\sum_{i=0}^N (1-a_i)}{N}$, where $N$ is the number of sentences in candidate $y^i$. Then, we construct a preference dataset based on factuality scores, where we assign $y^i$ with higher scores to $y_w$ and the lower ones to $y_l$. Finally, we filter the data to filter out preference pairs where $y_w$ and $y_l$ are the same, $y_w$ does not contain correct facts, and $y_l$ does not contain incorrect facts.

**Fine-grained Preference Learning.** Using the constructed fine-grained preference dataset $\mathcal{D} = \{x, y_w, s_w, a_w, y_l, s_l, a_l\}$, Mask-DPO forces the model to only learn from the correct facts in the preferred examples $y_w$ and the incorrect contents in the un-preferred examples $y_l$. Based on Eq. 3, we designed a masking scheme applied to the DPO algorithm as:

$$\begin{aligned}
\mathcal{M}_w(x, s_w, a_w) &= \sum_i \mathbb{I}(a_i = 0) \log \frac{\pi_\theta(s_i|x, s_{1\sim i-1})}{\pi_{\text{ref}}(s_i|x, s_{1\sim i-1})} \\
\mathcal{M}_l(x, s_w, a_w) &= \sum_j \mathbb{I}(a_j = 1) \log \frac{\pi_\theta(s_j|x, s_{j\sim j-1})}{\pi_{\text{ref}}(s_j|x, s_{1\sim j-1})}
\end{aligned} \tag{5}$$

where $\mathcal{M}_w$ is the masked KL divergence for $y_w$ while $\mathcal{M}_l$ for $y_l$. Combining Eq. 2 and Eq. 5, we formulate the optimization objective of fine-grained factuality tuning:

$$\mathcal{L}_{\text{Mask-DPO}}(\pi_\theta; \pi_{\text{ref}}) = -\mathbb{E}_{(x,s_w,a_w,s_l,a_l)\sim\mathcal{D}} \left[ \log \sigma \left( \beta \mathcal{M}_w(x, s_w, a_w) - \beta \mathcal{M}_l(x, s_l, a_l) \right) \right] \tag{6}$$

Table 1: **Evaluation results for the open-source models, FactTune, and our Mask-DPO.** Here, ANAH and Biography represent the in-domain and out-of-domain test sets, respectively. ANAH-v2 and FactScore represent the corresponding evaluation strategies. For each evaluation strategy, we report the number of correct facts (# Correct.), the number of inaccurate facts (# Inc.), and the factuality score (% Score) , *i.e.* the proportion of correct facts out of the total number of facts. **Bold** and underlined represent the best and second best performance, respectively.

| Model | ANAH (in-domain) | | | | | | Biography (out-of-domain) | | |
| | ANAH-v2 Evaluator | | | Factscore Evaluator | | | Factscore Evaluator | | |
| | # Cor. | # Inc. | % Score ↑ | # Cor. | # Inc. | % Score ↑ | # Cor. | # Inc. | % Score ↑ |
| --- | --- | --- | --- | --- | --- | --- | --- | --- | --- |
| Qwen2-7B | 450 | 872 | 34.03 | 5.19 | 30.60 | 15.57 | 14.46 | 39.27 | 27.28 |
| Qwen2-72B | 538 | 690 | 43.81 | 6.45 | 27.97 | 20.76 | 20.13 | 40.09 | 34.06 |
| Gemma2-9B | 635 | 837 | 43.13 | 4.51 | 25.92 | 18.29 | 15.53 | 29.00 | 29.52 |
| Gemma2-27B | 889 | 1110 | 44.47 | 7.81 | 35.65 | 20.14 | 19.27 | 36.59 | 29.99 |
| Yi1.5-6B | 440 | 1143 | 27.79 | 6.01 | 44.48 | 11.92 | 10.84 | 59.65 | 15.81 |
| Yi1.5-9B | 483 | 1136 | 29.83 | 5.06 | 39.79 | 10.42 | 10.63 | 62.67 | 15.33 |
| Yi1.5-34B | 535 | 911 | 36.99 | 5.25 | 36.96 | 13.27 | 17.01 | 52.76 | 25.49 |
| Llama3.1-8B | 461 | 476 | 49.19 | 5.95 | 21.29 | 19.43 | 16.83 | 31.57 | 30.29 |
| Llama3.1-70B | 520 | 453 | 53.44 | 6.81 | 22.17 | 21.92 | 23.58 | 31.59 | **40.47** |
| FactTune | 657 | 499 | 56.83 | 7.45 | 23.08 | 22.67 | 15.93 | 23.97 | 37.97 |
| Ours | 547.6 | 161.4 | **77.53** | 5.90 | 18.59 | **25.56** | 12.16 | 15.24 | 39.39 |

## 3 EXPERIMENT

### 3.1 EXPERIMENT SETUP

**Implementation.** In our experimental framework, we adopt the Llama3.1-8B-Instruct (Dubey et al., 2024) as the base model and the ANAH-v2 (Gu et al., 2024) as the fine-grained reward model to annotate factuality. All the experiments are conducted using the same setting for a fair comparison. Further implementation details can be found in Appendix A.

**Dataset.** For the in-domain data, we use a subset of the ANAH-v2 (Gu et al., 2024) data as the test set, containing 177 questions, and another 8046 questions as the training set. ANAH is organized along two dimensions: topics and questions, and the training set has no overlap with the test set in both dimensions. For the out-of-domain data, we use a subset of Biography (Min et al., 2023) as the test set, which has 183 questions about biography generation and has no overlap with our training set.

**Evaluation.** To evaluate each generated response, we use counts of correct and incorrect facts computed by ANAH-v2 (Gu et al., 2024) and FactScore (Min et al., 2023) as the evaluation metrics. ANAH-v2, which was trained specifically on our in-domain data for the hallucination annotation task, would annotate each sentence in the response with hallucinations, and we used the ratio of non-hallucinated sentences to the total number of sentences as the final score. Since ANAH-v2 has been used in preference data construction, to avoid reward hacking, we also use FactScore for evaluation, which calculates the number of correct facts and reports its proportion. In addition, considering that ANAH-v2 has not yet been trained in the task of generating biographies, and to ensure the reliability of the results, we only use FactScore as an evaluation strategy when evaluating out-of-domain data. For stability, we default to reporting the mean value after five replications.

**Baseline.** We construct the baseline in two ways. First, we compare Mask-DPO with many open-source models, including Qwen2 (7B & 72B) (Yang et al., 2024a), Gemma2 (9B & 27B) (Team et al., 2024), Yi1.5 (6B & 9B & 34B) (Young et al., 2024), and Llama3.1 (8B & 70B) (Dubey et al., 2024). Note that all open-source models used here are post-trained versions and we only report results from a single experiment for them. In addition, we also compare FactTune (Tian et al., 2023), a hallucination mitigation method that is also based on DPO. We apply it to the same base model.

### 3.2 OVERALL RESULTS

**Evaluation on In-Domain Data.** The first six columns of Tabel 1 illustrate the performance of the open source models, FactTune, and our Mask-DPO on the in-domain data. Under the ANAH-v2 evaluation strategy, Mask-DPO achieves the highest factuality score (77.53%), surpassing the best

Table 2: **Ablation study for the masking scheme applied to the DPO.** Here, "w/ mask" means that the factual alignment algorithm is DPO with mask, which is the default setting of Mask-DPO. "w/o mask" means that the algorithm is the vanilla DPO.

| Training Setting | ANAH-v2 | | | FactScore | | |
|---|---|---|---|---|---|---|
| | # Correct | # Incorrect | % Score ↑ | # Correct | # Incorrect | % Score ↑ |
| w/o mask | 446.60 | 206.00 | 68.44 | 5.71 | 18.98 | 23.43 |
| w/ mask | 547.60 | 161.40 | **77.53** | 5.90 | 18.59 | **25.56** |

Table 3: **Ablation study for the sampling strategy in preference dataset construction.** Here, "Llama" means that the data used to construct preference pairs is sampled from the policy model (Llama3.1-8B-Instruct), which is the default setting of Mask-DPO. "InternLM" means that the data is sampled from InternLM2-7B-Chat-SFT. And "Llama+Doc" means that the data is sampled from the policy model, but the sampling strategies for preferred and dispreferred samples are different. When generating preferred samples, the reference document corresponding to the question is added to the prompt, but not to the dispreferred ones.

| Sampling Setting | ANAH-v2 | | | FactScore | | |
|---|---|---|---|---|---|---|
| | # Correct | # Incorrect | % Score ↑ | # Correct | # Incorrect | % Score ↑ |
| InternLM | 419.20 | 296.80 | 59.43 | 5.19 | 16.59 | 21.33 |
| Llama+Doc | 877.00 | 1347.00 | 39.43 | 7.98 | 33.07 | 16.16 |
| Llama | 547.60 | 161.40 | **77.53** | 5.00 | 18.59 | **25.56** |

open-source model Llama3.1-70B-Instruct (53.44%) and factuality alignment method FactTune (56.83%). In addition, under the FactScore evaluation strategy, the results also show the same trend, where Mask-DPO reaches the highest factuality score (25.56%), surpassing the best open-source model Llama3.1-70B-Instruct (21.92%) and factuality alignment method FactTune (22.67%). These results also show that our reward model has not been overly and seriously hacked.

Tabel 1 also provides the factuality levels of existing open-source models, among which the Llama3.1 series achieves the highest factuality level, while the Yi1.5 series has the lowest level. Moreover, as the scale of model parameters increases, the factuality level of the model also increases, which is consistent with the observations of Gu et al. (2024).

**Evaluation on Out-of-Domain Data.** The last three columns of Tabel 1 illustrate the performance of the open source model, FactTune, and our Mask-DPO on the out-of-domain data. Without using the corresponding training set, Mask-DPO still exhibits excellent results, improving the factuality score of Llama3.1-8B-Instruct on the Biography dataset from 30.29% to 39.39%, surpassing the factuality alignment method FactTune (37.97%) and reaching a level close to the best open-source model Llama3.1-70B-Instruct (40.47%).

Moreover, the trends of the factuality levels between different series of open-source models and between different parameter amounts in the same series are consistent with those in Table 1, further illustrating the reliability of our results.

## 3.3 ABLATION STUDY

**Impact of Mask Scheme.** To verify the effectiveness of the masking scheme (introduced in § 2.2), we compare the performance of the models trained with and without the masking scheme. As shown in Tabel 2, the scheme with the mask is significantly better than the scheme without the mask in both evaluation strategies, which demonstrates the effectiveness of Mask-DPO.

It is worth noting that FactTune (the second to last row in Table 1) also uses the vanilla DPO for factuality alignment, but its final performance is not as good as the vanilla DPO in this experiment (the first row in Table 2). The main difference between them is that FactTune uses FactScore to construct preference data, while the vanilla DPO in this experiment uses ANAH-v2. Because FactScore and ANAH-v2 are also our two evaluation models, this result is shown in both evaluation strategies, which rule out the possibility of reward hackers. Therefore, we believe that the reward model used to construct preference data will have an obvious impact on the effectiveness of factuality alignment.

Table 4: **Evaluation for the efficiency and the generalization effects of different training sample scaling strategies.** Here, "Topic" means scaling from the number of different topics, and "Question" means scaling from the number of different questions under the same topic. " Topic" denotes the number of topics used for training, and " Question" denotes the number of questions under the same topic used for training. We also report changes in scores (in parentheses) when scaling the data.

| Dimension | # Topic | # Question | Scale | # Correct | # Incorrect | % Score ↑ |
|---|---|---|---|---|---|---|
| Topic | 894 | 3 | 1/3 | 501.80 | 270.20 | 65.04 |
| | 1788 | 3 | 2/3 | 518.00 | 206.20 | 71.56 (+6.52) |
| | 2682 | 3 | 1 | 547.60 | 161.40 | 77.53 (+5.97) |
| Question | 2682 | 1 | 1/3 | 486.00 | 218.00 | 69.03 |
| | 2682 | 2 | 2/3 | 444.00 | 164.00 | 73.03 (+4.00) |
| | 2682 | 3 | 1 | 547.60 | 161.40 | 77.53 (+4.50) |

**Impact of Sampling Strategy.** To assess the impact of the data sampling strategy (introduced in § 2.2), we compare the performance of the models trained with different data sampling strategies in Tabel 3. In our default setting, we sample the data from the policy model (Llama) to construct the factual preference dataset. We also test two other different sampling strategies, sampling from the non-policy model (InternLM) and sampling from the policy model but using different contexts for preferred and non-preferred samples (Llama + Doc).

As shown in Tabel 3, the performance of sampling from the non-policy model is far less effective than sampling from the policy model, even though it can improve on the base model. Notably, for settings that sample from the policy model but use different contexts for preferred and non-preferred samples, the scores after factuality alignment even drop from the base model. Intuitively, the quality of preference pairs constructed in this setting should be higher due to the presence of corresponding reference documents in the prompt while generating the preferred samples. This may be the case because the prompts used to generate the preferred and non-preferred samples are not the same, resulting in a negative impact on the factuality alignment although the quality of the preference pairs is higher. This phenomenon is also observed by Lin et al. (2024).

## 4    ANALYSIS OF DATA SCALING

We further analyze the efficiency and the generalization effects of different training sample scaling strategies of Mask-DPO (§ 4.1). To explain the observed phenomenon of generalization, we provide a hypothesis that discusses how factuality alignment implicitly impacts the internal knowledge structure in LLMs (§ 4.2), and then briefly design a series of proof-of-concept experiments to verify it. (§ 4.3).

### 4.1    ANALYSIS

In our training set, data are organized along two dimensions: topics and questions, where each question can be categorized into a topic, which is usually the entities the question asks about. Both topics and questions in the training set have no overlap with that of the test set. We analyze the impact of scaling the training data through these two dimensions, including the number of different topics and the diversity of questions under the same topic, respectively.

In the default setting of Mask-DPO, we use 2682 topics for factuality alignment, where each topic contains 3 questions. In the scaling of the number of topics, we use one-third, two-thirds, and the full number of topics for alignment. In the question dimension, we use the same scaling strategy, *i.e.*, only use one, two, or three questions per topic for alignment. Since we have demonstrated the consistency of ANAH-v2 and FactScore in terms of results in the previous section (§ 3), for resource considerations, we only use ANAH-v2 for the evaluation afterward.

As shown in Table 4, we observe that scaling the number of topics is more effective than increasing the diversity of questions. When scaling the same amount of data, scaling the topic leads to larger increases (6.53 and 5.97) compared to scaling the question (4.00 and 4.50). Moreover, the scores for the "scaling number of questions" setting are consistently higher than those for the "scaling number of topics" with the same amount of training data. Since the former consistently covers the full amount of topics, it suggests that the diversity of topics is more important than the diversity of questions.

## 4.2 HYPOTHESIS

The experimental results in previous sections indicate that after factuality alignment with questions from a group of topics, the LLM will become more factually correct when answering questions from another group of topics. On the implication of this phenomenon, we hypothesize that the factuality alignment essentially adjusts a model-specific graph consisting of topics as nodes represented by the LLM, where preference learning on some of the topics (nodes in the graph) will be propagated to other nodes through the graph.

Specifically, consider a graph $G_\theta = (V, E)$, where $V$ is the set of nodes, each representing a distinct topic and their corresponding knowledge, and $E \subseteq V \times V$ is the set of edges, representing relationships between topics. The edges are defined by the affinity between these topics, which are learned specifically for each model through the language modeling objective in training (especially pre-training). Thus, the hallucination can be interpreted as the blurring of knowledge between a node and other nodes. For any two topics $u, v \in V$, given the user query $x$, the hallucination phenomenon can be formulated as:

$$h(x, v \to u) \implies s_i = \pi_\theta \left( \cdot \mid x, s_{1 \sim i-1} \right), \quad x, s_{1 \sim i-1} \in v, \ s_i \in u, \tag{7}$$

where $s_{1 \sim i-1}$ represents the first $i - 1$ sentences in the reply of LLM, related to topic $v$, and $s_i$ represents the $i^{th}$ sentence, which erroneously shifts to information of topic $u$ due to the boundary blurring between topics $u$ and $v$.

Let an embedding function $\phi : V \to \mathbb{R}^d$ maps each topic $v \in V$ to a high-dimensional vector $\phi(v)$ in the embedding space $\mathbb{R}^d$. Due to the duality of the graph structure, the relationship between $v$ and $u$ can be quantified by a distance metric $D$, which we hypothesize can be approximated by the probability of its hallucination phenomenon:

$$\frac{1}{D(\phi(u), \phi(v))} \propto \mathcal{P}(h(x, u \to v)) \approx \mathcal{P}(h(x, v \to u)) = \pi_\theta \left( s_i \mid x, s_{1 \sim i-1} \right), x, s_{1 \sim i-1} \in v, \ s_i \in u \tag{8}$$

We hypothesize that more related topics will have smaller distances in the embedding space represented by LLMs, and thus are more likely to exhibit hallucination. This also explains that $s_i$ often behaves plausibly and is hard to detect (Ji et al., 2022) due to the proximity of $u$ and $v$.

With this formulation, factuality alignment using Eq. 2 and Eq. 3 on topic $v$ can be considered as decreasing the probability of $h(x, v \to u)$. Due to the duality of the affinity between nodes (Eq. 8), nearby topics like $u$ are implicitly adjusted together, which explains the generalization phenomenon. Moreover, Mask-DPO introduces a more accurate adjustment on each topic using Eq. 5 and Eq. 6, which makes the factuality alignment more effective.

## 4.3 PROOF OF CONCEPT

If the model-specific graph does exist, and our interpretation of factuality alignment on the graph is correct, then there will be two corollaries: 1) when factuality alignment is performed on a topic, topics closer to it will be more affected, as evidenced by the factuality score of that topic. 2) factuality alignment does not simply change the generation probability distribution or the style of certain content but has modified the internal knowledge structure of LLMs, which can be observed through best-of-N sampling.

**Proof of Corollaries (1).** To assess the impact between topics, we design an experiment based on topic clustering. We use different embedding strategies to get the high-dimensional vector of each topic, including embeddings using the policy model itself, embeddings from other models, and embeddings from specialized word embedding models. Then, we cluster the topics in the training set using the topic vectors in the test set as cluster centers, and split the training set into two groups, namely the far group and the near group, according to the distance from the corresponding topic vector to the cluster center. We also use a random strategy to select two groups as baselines. Finally, we use the far group and the near group for factual alignment under the same settings.

We conduct experiments using Llama3.1-8B-Instruct (Dubey et al., 2024) and InternLM2-7B-SFT-Chat (Cai et al., 2024) as the base (policy) models and use ANAH-v2 for evaluation.

Table 5: **Evaluation for the effects of different clustering strategies for preference data construction.** Here, "InternLM2-7B", "Llama3.1-8B" and "OpenAI" denotes InternLM2-7B-Chat-SFT, Llama3.1-8B-Instruct and Text-Embedding-3-Large, respectively. And "Random" means randomly selecting samples. "Distance" represents the distance between the training set and the cluster center. "Diff" represents the difference between the score at near distance settings and the one at far.

| Base Model | Embedding Model | Distance | # Correct | # Incorrect | % Score ↑ | Diff ↑ |
|---|---|---|---|---|---|---|
| InternLM2-7B | InternLM2-7B | far | 493.10 | 267.60 | 64.99 | **2.43** |
| | | near | 456.40 | 222.30 | **67.42** | |
| | Llama3.1-8B | far | 454.80 | 252.20 | 64.32 | 0.65 |
| | | near | 487.00 | 263.00 | 64.97 | |
| | OpenAI | far | 487.64 | 256.09 | 65.59 | 1.25 |
| | | near | 489.50 | 242.70 | 66.84 | |
| | Random | - | 453.60 | 247.40 | 64.78 | 0.77 |
| | | - | 521.70 | 275.90 | 65.55 | |
| Llama3.1-8B | InternLM2-7B | far | 455.50 | 206.50 | 68.97 | 0.22 |
| | | near | 551.75 | 280.75 | 69.19 | |
| | Llama3.1-8B | far | 568.00 | 265.75 | 68.94 | **4.49** |
| | | near | 500.00 | 182.33 | **73.43** | |
| | OpenAI | far | 761.25 | 317.00 | 71.32 | 0.85 |
| | | near | 616.60 | 253.60 | 72.17 | |
| | Random | - | 566.00 | 219.00 | 70.60 | 0.56 |
| | | - | 454.00 | 182.33 | 71.11 | |

Table 6: **Evaluation for the performance under the setting of best-of-N.** Here, "Baseline" represents the base model Llama3.1-8N-Instruct and "Mask-DPO" represents the aligned model. [1]

| Model | Best of 1 | Best of 16 | Best of 32 | Best of 64 | Best of 128 | Best of 256 |
|---|---|---|---|---|---|---|
| Baseline | 47.79 | 73.38 | 76.30 | 76.93 | 79.42 | 80.98 |
| Mask-DPO | 70.67 | 89.80 | 90.53 | 92.81 | 93.47 | 94.09 |

As shown in Table 5, regardless of the embedding strategy used, the factual score of alignment using the near group is higher than that using the far group. This indicates when factuality alignment is performed on a node, it affects other nodes, and the effect becomes smaller as the distance between nodes increases. Furthermore, when the base model and the embedding model are the same, the score difference between the far and near groups is the largest, *i.e.*, 2.43 for InternLM2-7B and 4.49 for Llama3.1-8B, and the near group obtains the best results (67.2% for InternLM2-7B and 73.43% for Llama3.1-8B), while the score differences for the other strategies are close to those for the random strategy and the near group chose by other strategies cannot obtain the highest results. This suggests that the graph-like knowledge structure is model-specific.

**Proof of Corollaries (2).** We further test the best-of-N performance of the Llama3.1-8B-Instruct (Dubey et al., 2024) before and after training, using ANAH-v2 for evaluation. Specifically, we perform top-k (k = 40) sampling in inference time, and chose the most factual response from N candidates as the final response for each question.

As shown in Table 6, the best-of-N scores for both settings rise in parallel as the number of samples increases. However, the difference in scores before and after factuality alignment is always present and does not gradually converge as the number of samples increases. This is inconsistent with what Havrilla et al. (2024) observed in the math reasoning task, where the scores before and after training would converge as the number of samples increased, which means that the training only changes the probability distribution that generates the correct answer. Such a phenomenon proves our corollaries that factuality alignment has an essential effect on the knowledge structure within LLMs and does not simply change the generation probability distribution of certain content.

---

[1]The results for best-of-1 do not match those in Table 1 because the inference setting here is top-k=40, whereas in Table 1 it is top-k=1.

## 5 RELATED WORK

**Reinforcement Learning from Human Feedback.** In order to unlock the capabilities of LLMs and align them with human preferences, a bunch of work about alignment has been proposed. Reinforcement Learning from Human Feedback (RLHF) (Ouyang et al., 2022; Bai et al., 2022; Rafailov et al., 2024; Yuan et al., 2024) is the typical alignment method, a representative work is Proximal Policy Optimization (PPO) (Schulman et al., 2017). PPO uses paired data based on human preferences to train a reward model and then uses the signals provided by that reward model to guide the optimization of the policy model, which is complex. For simplicity, Direct Preference Optimization (DPO) (Rafailov et al., 2024) utilizes pairwise preference data directly for model optimization. Since using humans for preference data construction is costly (Min et al., 2023), Reinforcement Learning from AI Feedback (RLAIF) (Lee et al., 2023) is proposed, which constructs preference data using AI (*e.g.* advanced LLMs, specialized evaluator models, and agents system) feedback. Our approach can be regarded as a fine-grained RLAIF method for hallucination mitigation, which uses feedback from the hallucination annotator as the reward signal and we improve DPO for more effective factuality alignment.

**Hallucination Mitigation.** Considering the harm of hallucinations, researchers have explored various techniques for enhancing the factuality of LLMs. To evaluate the factuality of LLMs, researchers have developed various benchmarks (Ji et al., 2024; Wei et al., 2024; Min et al., 2023) and hallucination detection methods (Gu et al., 2024; Min et al., 2023). However, these methods only detect issues without providing solutions. For mitigation, multi-task learning (Garg et al., 2019; Weng et al., 2020), model editing (Daheim et al., 2023; Ji et al., 2023), and factuality alignment (Wu et al., 2023; Tian et al., 2023; Lin et al., 2024; Zhang et al., 2024b; Wen et al., 2024; Xu et al., 2024; Chen & Li, 2024; Jeong et al., 2024) are attempted. Some of them (Wu et al., 2023; Wen et al., 2024) perform fine-grained RLHF using PPO, while our approach is based on the more efficient and cost-friendly DPO. Most similar to ours, some approaches attempt to mitigate hallucination using DPO, and their main difference lies in the different ways of constructing preference data. For example, Lin et al. (2024); Jeong et al. (2024) uses an external estimator or annotator to construct factual preference data, while Chen & Li (2024), Zhang et al. (2024b) rely on the model itself, and Tian et al. (2023) has both solutions. However, all of them only use the factuality of the entire response as the sparse reward signal, making factuality alignment less effective. In contrast, our approach implements fine-grained factuality alignment using sentence-level reward signals.

**Knowledge Editing.** The primary cause of the hallucination is the fragility of parametric knowledge in LLMs (Wang et al., 2024a). To keep knowledge truthfulness and reliability, many works attempt to edit it, including knowledge and representation editing. Knowledge editing (Zhang et al., 2024a; Wang et al., 2024b; McAleese et al., 2024) focuses on selectively adjusting model parameters that store specific pieces of knowledge, while representation editing (Zou et al., 2023; Wu et al., 2024) modifies how the model conceptualizes knowledge to update the information retained within LLMs. Unlike them, our approach does not explicitly edit the knowledge-related weights or representations but we hypothesize factuality alignment implicitly modifies the knowledge structure inside the model.

## 6 CONCLUSION

In this paper, we introduce a fine-grained factuality alignment method, Mask-DPO. Incorporating sentence-level factuality as reward signals, Mask-DPO accurately masks the encouraging signal of incorrect sentences in the preferred samples and the punishment signal from factual contents in the non-preferred samples, thus improving the effectiveness of learning. Extensive experimental results demonstrate that Mask-DPO significantly increases the factuality of LLMs in both in-domain and out-of-domain evaluation. The study on the generalization property of Mask-DPO using different training sample scaling strategies reveals that the diversity of topics is more important to the generalization than the number of questions. On the implication of this observation, we hypothesize that LLM learns a model-specific graph-like knowledge structure in training, which propagates the impact of factuality alignment based on the distance between topics, where nearby topics will also be implicitly aligned, even if they are not seen during alignment. We conduct proof-of-concept experiments and show that factuality alignment indeed essentially aligns the internal knowledge of LLMs. We hope our study could pave the way for future research on scaling the factuality alignment of LLMs.

ACKNOWLEDGEMENT

We thank the anonymous reviewers and area chair for their helpful comments. This project is supported by the Shanghai Artificial Intelligence Laboratory. The authors would like to thank Songyang Gao and Kuikun Liu for their valuable suggestions and comments.

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

## A  IMPLEMENTATION DETAILS

In our experimental framework, we adopt the Llama3.1-8B-Instruct (Dubey et al., 2024) as the base model and the ANAH-v2 (Gu et al., 2024) as the fine-grained reward model to annotate factuality.

**Preference data construction.** We generate 32 candidate responses per question and selected the ones with the highest and lowest scores to form preference pairs after fine-grained factuality estimation. The decoding strategy involves the top-k (k = 40) sampling with a temperature of 0.8.

For each question, we sample multiple responses from the model. For a response, we first split it into several sentences, and then annotate them sentence by sentence using the hallucination annotator. We choose ANAH-v2 (Gu et al., 2024) as the annotator, which provides four types of labels for each sentence, i.e., 'No Facts', 'No Hallucination', 'Contradictory' and 'Unverifiable'. Following the same setup as in ANAH-v2, we treat the first two types as hallucination-free and the last two as hallucination. We take the proportion of sentences in a response that is hallucination-free as the factuality score of that response. Based on the factuality score, we rank between multiple responses and select the highest-scoring and lowest-scoring responses to form preference pairs.

**Fine-grained factuality tuning.** We train the base model with the following settings and hyper-parameters: the epoch is 3, the learning rate is 5e-6, the batch size is 64, and the AdamW optimizer is the cosine annealing learning rate scheduler.

Our model is trained on 8 NVIDIA A100 GPUs. The training and inference frameworks are Xtuner (Contributors, 2023b) and LMDeploy (Contributors, 2023a) respectively.

**Embedding, clustering, and choosing the far group and the near group.** For each topic, we use the output of the last hidden layer of the model as its embedding vector. The embedding vectors of the topics in the test set are treated as multiple cluster centroids. For each topic in the training set, we calculate the Euclidean distance from its embedding vector to each cluster centroid. Each training set topic is then assigned to the closest cluster centroid, forming a series of clusters. Within each cluster, the training set topics are further divided into two groups—far and near—based on their distances to the cluster centroid. Topics closer to the centroid are grouped into the near group, while those farther away are grouped into the far group. Finally, the far groups from all clusters are merged into a single large far group, and similarly, the near groups are merged into a single near group. Based on the group to which a topic belongs, preferred pairs are reorganized into two separate training datasets. These datasets are then trained separately.

**Prompt.** We do not use any special prompts other than the hallucination annotations. Both in the construction of the preference data and in the evaluation process, we directly use the question in the dataset as a prompt. For hallucination annotation in fine-grained preference data construction, we use the prompt from Gu et al. (2024), including factual existence judgment, reference information extraction, and hallucination-type judgment.

## B  ADDITIONAL EXPERIMENTS

In this section, we provide additional experiments to further validate the effectiveness of Mask-DPO, including comparisons with other methods (§ B.1), evaluations on standard benchmarks (§ B.2), and the impact of different inference hyper-parameter (§ B.3).

## B.1 COMPARATIVE EXPERIMENTS

**Comparision with SFT+PO.** The default setting of Mask-DPO is to perform fine-grained preference learning on the instruct model. To provide greater transparency on the training procedure, we compare it with performing SFT+PO on the base model. Specifically, we apply SFT to the base model (not instruction-tuned), Llama3.1-8B, using the chosen responses from the existing preference data. Subsequently, we perform Mask-DPO on the supervised model.

As shown in Table 7, executing SFT+Mask-DPO from the base model has similar results as executing Mask-DPO directly from the struct model, which shows that our fine-grained preference learning approach does not have any specific assumption or reliance on the SFT process.

Table 7: Comparision with SFT+PO.

|  | # Correct | # Incorrect | % Score |
|---|---|---|---|
| Llama-3.1-8B + SFT + Mask-DPO | 631 | 192 | 76.67 |
| Llama-3.1-8B-Instruct + Mask-DPO | 547.6 | 161.4 | 77.53 |

**Comparision with other PO Strategies.** We conduct comparative experiments on TDPO (Zeng et al., 2024), DPOP (Pal et al., 2024), and SePO (Yang et al., 2024b).

The specific configurations are as follows: (1) TDPO: We use the default settings from their implementation, specifically the TDPO2 method with the alpha parameter set to 0.5. (2) DPOP: The lambda parameter is set to 50, consistent with the original paper. (3) SePO: We utilize the Oracle Model provided in their work to estimate the token-level reward function. For all experiments, we use Llama3.1-8B-Instruct as the base model and ANAH-v2 for evaluation.

As shown in Table 8, the performance of TDPO and DPOP is similar to DPO, while SePO performs significantly worse than DPO. This discrepancy may be due to the Oracle Model used in SePO not being trained on our tasks, which limits its ability to accurately estimate the token-level reward.

Table 8: Comparision with other PO Strategies.

|  | # Correct | # Incorrect | % Score |
|---|---|---|---|
| DPO | 446.4 | 206 | 68.44 |
| DPOP | 593 | 291 | 67.08 |
| TDPO | 703.6 | 284 | 70.95 |
| SePO | 254 | 611.4 | 58.45 |
| Mask-DPO | 547.6 | 161.4 | 77.53 |

**Comparision with factuality alignment methods.** To help better position Mask-DPO in the context of the existing approach, except FactTune (Tian et al., 2023), we supplement the comparison with another state-of-the-art factuality alignment method, i.e., Flame (Lin et al., 2024). Specifically, we construct factuality preference data on our training set and perform SFT+DPO following the method described in Flame. We use Llama3.1-8B-Instruct as the base model and use ANAH-v2 for evaluation. Table 9 reports the comparison between Mask-DPO and the other two SOTA methods, where Mask-DPO gives better results.

Table 9: Comparison with factuality alignment methods.

|  | # Correct | # Incorrect | % Score |
|---|---|---|---|
| FactTune | 657 | 499 | 56.83 |
| Flame | 447 | 263 | 62.96 |
| Mask-DPO | 547.6 | 161.4 | 77.53 |

## B.2 EVALUATION ON STANDARD BENCHMARKS

To analyze the effect of factuality alignment over other LLM capabilities, we supplement the evaluation with relevant benchmarks for math and code. Specifically, we test the model on the GSM8K, MATH, GPQA, Human Eval, and MBPP datasets using the OpenCompass evaluation framework. The base model is Llama3.1-8B-Instruct.

Table 10 presents the results of the model on these benchmarks before and after training. Mask-DPO shows a slight decrease in performance on one math benchmark (MATH) and two code benchmarks (Human Eval and MBPP), while demonstrating improvements on two math benchmarks (GSM8K and GPQA). These results suggest that factuality alignment methods may have marginal effects on the other capabilities of the model.

Table 10: Evaluation on Standard Benchmarks.

|          | GSM8K | MATH  | GPQA  | Human Eval | MBPP  |
|----------|-------|-------|-------|------------|-------|
| Baseline | 84.91 | 52.72 | 26.77 | 70.73      | 71.21 |
| Mask-DPO | 86.20 | 47.68 | 31.82 | 68.29      | 70.04 |

## B.3 IMPACT OF DIFFERENT INFERENCE HYPER-PARAMETER

To analyze the effect of different inference hyper-parameters, i.e. temperature, we conduct a series of experiments to investigate the effect of this parameter on performance. Specifically, we tested four temperature settings: 0.25, 0.5, 0.75, and 1.0. We exclude 0 because sampling at this value produced insufficiently diverse data for our task, making it challenging to construct preference data. The other experimental conditions are consistent with those described in the paper. We use Llama3.1-8B-Instruct as the base model and evaluate performance using ANAH-v2.

The comparison results are shown in the Table 11. Besides the default setting (0.8), the best performance is observed at temperature = 0.75, while performance at temperature = 0.25 is lower. Interestingly, this trend aligns with findings in the DPO paper (Rafailov et al., 2024), despite differences in task objectives. This suggests that the optimal sampling temperature for our task is around 0.8.

Table 11: Evaluation on different inference setup.

| Temperature   | # Correct | # Incorrect | % Score |
|---------------|-----------|-------------|---------|
| 0.25          | 485       | 259.2       | 65.30   |
| 0.5           | 539       | 244.8       | 67.96   |
| 0.75          | 733       | 219         | 76.02   |
| 1.0           | 732       | 240.6       | 74.24   |
| 0.8 (default) | 547.6     | 161.4       | 77.53   |

## C ADDITIONAL DISCUSSION

In this section, we provide additional discussions about the hypothesis (§ C.1) we mentioned in § 4.2 and the impact of Mask-DPO on the quality of the response (§ C.2).

### C.1 DISCUSSION ABOUT HYPOTHESIS

**Affinity analysis.** The idea-there is some structure in language modeling-has been explored in previous classic works (Mikolov et al., 2013; Mikolov, 2013; Bengio et al., 2000), which argue that language models produce similar representations of words with high affinities. In other words, for an ideal language model, the relevance of the semantic space and the similarity of the semantic space represent spaces that are consistent.

Building on this understanding, we propose the internal knowledge structure hypothesis. As discussed in Section 4.2, we hypothesize that the model's blurred recognition of similar topics leads to the mixed use of their respective subordinate content, resulting in phenomena akin to hallucination (cf. Eq 8). We conceptualize factual alignment as a process of adjusting the relationships between topics. During the alignment of a given topic, the affinity between the topic and its subordinate content increases, while the affinity with the subordinate content of similar topics decreases. This ultimately manifests as a reduction in hallucinated content.

We randomly selected 200 data from the preference dataset used for training to perform a quantitative analysis of affinities. Specifically, for each preference data point, we extracted the entities from both the chosen response and the rejected response, embedding these entities using word embeddings. We then calculated the average distance between the word vectors of these entities and the vector of the topic to which the preference data belongs. This evaluation was performed on the model separately before and after training. The results indicate that, after training, the average distance between the topic vector and the entities in the chosen response decreased by **1.97**, while the average distance to the entities in the rejected response increased by **4.09**. These findings support the reliability of our affinity-based explanation of the internal knowledge structure.

**Discussion about the question-based knowledge structure.** For the hypothesis about the internal knowledge structure, while Corollary 1 in Section 4.3 suggests that the structure may be topic-based, it does not rule out the possibility of a question-based structure. To address this, we designed an additional experiment inspired by Corollary 1's setup, focusing on clustering based on questions. Using Llama3.1-8B-Instruct as the base and embedding model for evaluation, we analyzed the results.

As shown in Table 12, the clustering outcomes based on questions and topics are not significantly different, indicating that the knowledge structure is not question-based.

Table 12: Evaluation on question-based clustering strategies for preference data construction.

| Distance | # Correct | # Incorrect | % Score | Diff |
|----------|-----------|-------------|---------|------|
| far      | 541.4     | 214         | 71.69   | 0.23 |
| near     | 518       | 195         | 71.92   |      |

### C.2    DISCUSSION ABOUT THE QUALITY OF THE RESPONSE

In order to analyze whether Mask-DPO affects the quality of the model-generated responses, e.g. overall consistency and completeness, we add a text quality evaluation. We use the GPT-2 to compute the average perplexity of text generated by the base model (Llama3.1-8B-Instruct) and the model after Mask-DPO training. Evaluated on the ANAH test set, the average PPL of the base model is **44.20**, while after Mask-DPO training the average PPL is **34.91**, suggesting that the training does not affect the quality of the generated text, and even makes it better.

## D    CASE STUDY

To analyze the effect of our method, we provide a case study about the generated response before and after Mask-DPO. As shown in Figure 3, there is a significant decrease in the number of phantom parts in the responses after Mask-DPO training, illustrating the effectiveness of our method.

## E    LIMITATION

Although this study presents a novel and effective fine-grained factuality alignment framework, there are some limitations. Despite Mask-DPO's performance in fine hallucination relief is remarkable, it remains somewhat exaggerated. Because the model we used to construct the preference data is the same as the one used for the review, this could pose a potential risk of reward hacking. Although we introduced additional evaluation models to show that this hacking phenomenon is not significant, it still cannot be completely avoided. Furthermore, we only perform evaluation on ANAH and Biography. However, these datasets might not encompass the full spectrum of real-world scenarios

where hallucinations pose a problem. And this work primarily uses Llama3.1-8B-Instruct and InternLM2-7B-Chat-SFT as the backbone of the base model. Other different underlying models and different numbers of parameters are not explored. Lastly, sentence-level hallucination annotations are not fine-grained enough. Notely, the form of Mask-DPO itself is general and does not assume that only sentence-level annotations can be used. The results in the paper show that using finer-grained annotations, such as sentence-level annotations, can lead to better results than using response-level annotations. If an external hallucination annotator can provide more detailed annotations, such as at the level of clauses, noun phrases, verb phrases, etc., we believe the results will be better. However, there is currently a lack of hallucination annotators that can provide more fine-grained information.

**Question:**
What is a hei-tiki?

**Reference Document:**
Hei-tiki, small neck pendant in the form of a human fetus, used by the Māori of New Zealand as a fertility symbol. Usually carved of green nephrite or a jadelike stone called pounamu that is found along the western coast of the South Island, hei-tikis normally are worn only by women. The object is believed to possess magical powers that increase as it is passed on from generation to generation. According to one idea, the hei-tiki protects its wearer against the vengeful spirits of stillborn infants, who have been deprived of the chance to live. Another theory holds that the figure represents the Polynesian god Tiki, the creator of life. Hei-tikis have been prized by European and American collectors for their beauty and elegance, but to the Māori the greatest value of these pendants lies in their possession of magical powers and in the prestige acquired from previous owners.

**Baseline:**
A hei-tiki is a traditional Māori pendant that originated in New Zealand. It is a significant cultural artifact that holds spiritual and historical importance for the indigenous Māori people.
A hei-tiki is a carved greenstone (pounamu) pendant in the shape of a human figure, typically a woman or a child. The pendant is usually made from a single piece of greenstone, which is a type of jade that is highly valued in Māori culture. The carving is often intricate and features a range of details, including facial features, hair, and clothing. Hei-tiki pendants were traditionally worn by Māori women. They were often passed down through generations as family heirlooms and were believed to possess spiritual energy and protection. The pendant was thought to bring good fortune, health, and prosperity to the wearer. In Māori culture, hei-tiki pendants were also associated with the concept of ẅhakapapa,ör genealogy. The pendant was seen as a connection to one's ancestors and the spiritual realm.
Today, hei-tiki pendants are highly valued for their cultural significance and are often worn as a symbol of Māori identity and pride. They are also highly sought after by collectors and are considered a significant part of New Zealand's cultural heritage.

**Mask-DPO:**
A hei-tiki is a traditional Māori pendant. It is a distinctive and iconic piece of Māori jewelry that is often associated with the Māori people's rich history and cultural heritage. Here are some key facts about hei-tiki:
1.**What is a hei-tiki?** A hei-tiki is a carved pendant that typically depicts a child figure.
2. **Significance:** Hei-tiki is considered a treasured and sacred object in Māori culture, often passed down through generations as a family heirloom.
3. **Purpose:** The exact purpose of hei-tiki is not well-documented, but it is believed to have been worn as a symbol of status, wealth.
4. **Cultural significance:** Hei-tiki is an important part of Māori cultural heritage and is often seen as a representation of the Māori people's rich history and traditions.
In summary, hei-tiki is a significant and sacred object in Māori culture, often associated with the people's history, traditions, and cultural heritage. I hope this helps you understand what a hei-tiki is!

Figure 3: **The case study about the generated responses before and after Mask-DPO.** Here, "Baseline" denotes the response from the model before Mask-DPO, i.e. Llama3.1-8B-Instruct. "Mask-DPO" denotes the response from the model after Mask-DPO. In generations, we use blue color to present the hallucination content.

