# OpenReview forum: "Mask-DPO: Generalizable Fine-grained Factuality Alignment of LLMs"
_ICLR.cc/2025/Conference — ICLR 2025 Poster_

### Official Review · Reviewer_AbTR · 2024-11-01

**Soundness:** 2
**Presentation:** 3
**Contribution:** 2
**Rating:** 6
**Confidence:** 4

**Summary:**

The paper presents a fine-grained factuality tuning method, referred to as Mask-DPO, designed to incorporate sentence-level factuality. Extensive experiments demonstrate the effectiveness on both in-domain and out-of-domain datasets. It also provides an analysis from two dimensions: the number of different topics and the diversity of the questions.

**Strengths:**

1.The approach of filtering out false sentences within preference pairs is particularly intriguing, as it helps refine the dataset by ensuring that only factual information is prioritized.

2.The paper establishes well-founded hypotheses regarding the functionality of the factual alignment process and conducts experiments to validate these assumptions.

**Weaknesses:**

1.Mask-DPO operates at the sentence-level granularity, employing a sentence-level hallucination annotator to mask undesired sentences within training preference pairs. However, it's worth noting that many responses mix true and false information in a single sentence. I'm curious how Mask-DPO handles these cases since pulling apart accurate details from inaccuracies can be tricky. Clarifying this might help us see how well the model deals with more complex responses.

2.The experiments seem focused on long-form generation, which raises the question of how the model would perform on shorter, QA-style tasks.

3.The proof of Corollary 2 in Section 4.3 is less convincing. How did you arrive at this knowledge structure corollary based on the hypothesis you mentioned earlier? Specifically, if the model-specific graph were focused on questions rather than topics, would we still arrive at the same conclusions? Please clarify this.

**Questions:**

Please see weakness.

Additionally, in Table 5, it’s intriguing that using embeddings from other models seems to make the differences less distinct. Why alternative embeddings are used instead of relying solely on those from the policy model itself?

I would like to increase my score if my concerns are addressed properly.

---

> ### Author Response · Authors · 2024-11-21
> **Responses to Official Review by Reviewer AbTR**
>
> Thank you for your constructive comment. The following are our responses to each individual comment (highlighted in italics).
>
> ### **Response to Weakness1 about potential errors due to sentence-level annotation:**
>
> > *Mask-DPO operates at the sentence-level granularity, employing a sentence-level hallucination annotator to mask undesired sentences within training preference pairs. However, it's worth noting that many responses mix true and false information in a single sentence. I'm curious how Mask-DPO handles these cases since pulling apart accurate details from inaccuracies can be tricky. Clarifying this might help us see how well the model deals with more complex responses.*
>
> We acknowledge the limitation of sentence-level hallucination annotations. When a sentence contains true or false information, we identify it as having a hallucination, which prevents learning the wrong information, although it also means that the correct information is lost.
>
> However, we believe that the bottleneck actually lies in the external hallucination annotator instead of our approach. The form of Mask-DPO itself is general and does not assume that only sentence-level annotations can be used. The results in the paper show that using finer-grained annotations, such as sentence-level annotations, can lead to better results than using response-level annotations.  If an external hallucination annotator can provide more detailed annotations, such as at the level of clauses, noun phrases, verb phrases, etc., we believe the results will be better. However, there is currently a lack of hallucination annotators that can provide more fine-grained information. Therefore, we believe the bottleneck lies with the external hallucination annotator, not our approach.
>
> We have added a corresponding discussion in the Limitation part of our revised manuscript (Appendix I, line 905-911).
>
> ### **Response to Weakness2 about the targeted task:**
>
> > *The experiments seem focused on long-form generation, which raises the question of how the model would perform on shorter, QA-style tasks.*
>
> We wish to clarify that we have not claimed that we are aiming at the long-form generation task. Our data contains both long and short-form QA data.
>
> ### **Response to Weakness3 about the proof of Corollary 2:**
>
> > *The proof of Corollary 2 in Section 4.3 is less convincing. How did you arrive at this knowledge structure corollary based on the hypothesis you mentioned earlier? Specifically, if the model-specific graph were focused on questions rather than topics, would we still arrive at the same conclusions? Please clarify this.*
>
> The purpose of Corollary 2 is to demonstrate a change in the structure of knowledge, but it does not determine whether this change is driven by questions or topics. While Corollary 1 suggests that the structure may be topic-based, it does not rule out the possibility of a question-based structure.
>
> To address this, we designed an additional experiment inspired by Corollary 1’s setup, focusing on clustering based on questions. Using Llama3.1-8B-Instruct as the base and embedding model for evaluation, we analyzed the results. As shown in the table below, the clustering outcomes based on questions and topics are not significantly different, indicating that the knowledge structure is not question-based.
>
> We have added a corresponding discussion and evaluation results in our revised manuscript (Appendix F, line 864-878).
>
> | Distance | #Correct | #Incorrect | %Score |
> | -------- | -------- | ---------- | ------ |
> | far      | 541.4    | 214        | 71.69  |
> | near     | 518      | 195        | 71.92  |
>
> ### **Response to Question about the setting of Table 5:**
>
> > *Additionally, in Table 5, it’s intriguing that using embeddings from other models seems to make the differences less distinct. Why alternative embeddings are used instead of relying solely on those from the policy model itself?*
>
> Using different embedding models is to demonstrate that the knowledge structure is model-specific. Specifically, according to the description in Section 4.2, if we accurately align the model using topics that are closer to the test set of topics on the knowledge structure graph, we will achieve better results than using topics that are farther away.
> Moreover, the model's embedding vector for a topic reflects its understanding of that topic, and a larger gap between results from alignment using near and far datasets indicates a more accurate understanding of the knowledge structure.
>
> Table 5 shows that significant divergence is observed only when the policy model is used as the embedding model, whereas other embedding models perform similarly to random selection. This confirms that only the policy model truly understands the underlying knowledge structure, emphasizing its model-specific nature.

---

> > ### Comment · Reviewer_AbTR · 2024-11-25
> >
> > Thanks for the response, which basically addresses my concerns. I will improve the rating score.

---

> > > ### Author Response · Authors · 2024-11-25
> > >
> > > Thank you for your response and for increasing the rating. We are happy that our discussions are convincing and address your concerns.

---

### Official Review · Reviewer_GYLY · 2024-11-01

**Soundness:** 3
**Presentation:** 3
**Contribution:** 3
**Rating:** 8
**Confidence:** 4

**Summary:**

This work presents a method for training models via preference optimization in order to produce more factually correct responses and as a result mitigate halluciantions. This is achieved by (1) creating preference data where chosen responses are more factually correct than dispreferred ones and (2) by masking parts of each response if they contain inacccurate information.
To realise the above, the authors use an external sentence-level factual evaluator to measure the correctness of each sentence inside each response and consider the ratio of correct sentences as the preference score. Then, a binary mask is employed into the DPO loss thus increasing the probability of correct facts and decreasing the probability of incorrect facts.

Results on in- and out-of-domain datasets suggest that the resulting models show significantly better performance across two different evaluators.
In addition, the authors hypothesize that such an alignment impacts the internal knowledge of LLMs, which they showcase via clustering responses on specific topics and best-of-N responses. Topics close to each other have better factuality.

**Strengths:**

- The authors propose a simple and straighforward method to improve models to produce more factually correct responses
- Results on the evaluated benchmarks indicate significant improvements
- Proof of concept experiments are designed to explain the performance of the models and the effect that factual alignment has in their internals

**Weaknesses:**

- One could argue that even in a single sentence, there can be factually incorrect information. For example, if the sentence is long enough, different clauses can contain conflicting information. How would you be able to mitigate this without completely ignoring the entire sentence?
- The method looks general enough, however it is targeted on factuality alone. Do you think if you replaced factual checking with cohesiveness / fluency / some other metric that is easy to obtain, models could also improve towards that direction too?

**Questions:**

Questions:
- Line 262: You mention that the models are post-trained versions, what do you mean by that? That the models are instruction tuned?
- I am kinda missing the point of the ablation in Table 3. Do you aim to show that the quality of responses depends on the underlying model used to generate them? In this setting, the data are generated by model X but LLaMA is trained with Mask-DPO? I think it is already proved that the model needs to be in-distribution for DPO to work (original DPO paper) - which explains why llama3 works best in this case too?

Comments:
- Line 219: inaccuracies facts -> incaccurate facts
- Line 283: un-preferred -> dispreferred (in other places as well)
- Line 323: rewarding hackers -> reward hacking
- Table 4: what does the +X refer to? which is the baseline score?
- I feel the name "Mask-DPO" is quite misleading as it is very generic and it is not directly associated with factuality.

---

> ### Author Response · Authors · 2024-11-21
> **Responses to Official Review by Reviewer GYLY**
>
> Thank you for your constructive comment. The following are our responses to each individual comment (highlighted in italics).
>
> ### **Response to Weakness1 about the sentence-level annotation:**
>
> > *If the sentence is long enough, different clauses can contain conflicting information. How would you be able to mitigate this without completely ignoring the entire sentence?*
>
> We acknowledge the limitation of sentence-level hallucination annotations. But we believe that the bottleneck actually lies in the external hallucination annotator instead of our approach.
>
> The form of Mask-DPO itself is general and does not assume that only sentence-level annotations can be used. The results in the paper show that using finer-grained annotations, such as sentence-level annotations, can lead to better results than using response-level annotations.  If an external hallucination annotator can provide more detailed annotations, such as at the level of clauses, noun phrases, verb phrases, etc., we believe the results will be better. However, there is currently a lack of hallucination annotators that can provide more fine-grained information. Therefore, we believe the bottleneck lies with the external hallucination annotator, not our approach.
>
> We have added a corresponding discussion in the Limitation part of our revised manuscript (Appendix I, line 905-911).
>
> ### **Response to Weakness2 about the other optimization directions:**
>
> > *The method looks general enough, however it is targeted on factuality alone. Do you think if you replaced factual checking with cohesiveness / fluency / some other metric that is easy to obtain, models could also improve towards that direction too?*
>
> Yes, I think your idea is reasonable. While this work focuses on factuality alignment, we believe Mask-DPO is a general approach, as it provides a more efficient way to utilize reward signals. The key lies in identifying a fine-grained reward model—such as the hallucination annotator used in this work—that can guide the optimization directions you mentioned. This model could then be effectively integrated with the Mask-DPO paradigm.
>
> ### **Response to Question about the unclear presentation:**
>
> > *1) Line 262: You mention that the models are post-trained versions, what do you mean by that? That the models are instruction tuned?*
> >
> > *2) I am kinda missing the point of the ablation in Table 3. Do you aim to show that the quality of responses depends on the underlying model used to generate them? In this setting, the data are generated by model X but LLaMA is trained with Mask-DPO? I think it is already proved that the model needs to be in-distribution for DPO to work (original DPO paper) - which explains why llama3 works best in this case too?*
>
> 1) Yes, your understanding is correct. We are using models that have been fine-tuned with instructions. For example, Llama3.1-8B in Table1 denotes the Llama3.1-8B-Instruct.
> 2) Your understanding is basically correct. However, we want to clarify that we are not proving that “the quality of responses depends on the underlying model used to generate them”. What we are trying to prove with our ablation experiments in Table3 is the validity of our preference data construction method, i.e. sampling from the policy model.  If we look at it from a quality point of view, the "Llama+Doc" setting in Table3, which generates higher quality data due to the introduction of RAG, should have been better, but it leads to significant performance degradation due to the different samping distributions. So what we want to show is that the effect of alignment depends on the method of constructing the preference data, and sampling from the same distribution in the policy model is the best. Indeed, as you said, this conclusion has been confirmed by other papers, but to rigorously prove the validity of our module, we still add this ablation study experiment.
>
> ### **Response to Comment about the typos and title:**
>
> > - *Line 219: inaccuracies facts -> inaccurate facts*
> > - *Line 283: un-preferred -> dispreferred (in other places as well)*
> > - *Line 323: rewarding hackers -> reward hacking*
> > - *Table 4: what does the +X refer to? which is the baseline score?*
> > - *I feel the name "Mask-DPO" is quite misleading as it is very generic and it is not directly associated with factuality.d*
>
> Thanks for your suggestion! We have fixed the typos you mentioned in our revised manuscript.
>
> In Table4, +X implies the amount of change in score from the score in the previous row as an indicator of sample efficiency. We notice a typo that +6.53 in the second row should be +6.52, and we have fixed it in our revised manuscript.
>
> For the title "Mask-DPO", we use it because it reflects our core approach well. We recognize that it does not have a strong association with factuality. But as we discussed with you earlier, we also think it is a generalized approach and we expect it to be useful for more tasks in the future.

---

> > ### Comment · Reviewer_GYLY · 2024-11-26
> > **Reviewer Response**
> >
> > Thank you for the answers and apologies for the late reply. I didn't have major concerns about the paper, hence I keep my score as is.
> > - For the ablation, thanks for the explanation, I think it would be beneficial to add to the paper "as has been proved by ..." to illustrate that you further support the claim with this experiment.
> > - Regarding the title, I still think it is misleading (although I agree with the above discussion) - perhaps you can consider adding something like "A case-study on factuality" or something similar, but no strong feelings.

---

> > > ### Author Response · Authors · 2024-11-28
> > >
> > > Thank you for your response. We are happy that our discussions are convincing. We will update the manuscript based on your comments in the next version.

---

### Official Review · Reviewer_upqN · 2024-11-02

**Soundness:** 3
**Presentation:** 3
**Contribution:** 3
**Rating:** 6
**Confidence:** 4

**Summary:**

It proposes Mask-DPO, a fine-grained factuality alignment method, which performs a sentence-level masking to denoise inaccurate training signals inherent from noisy sentences by utilizing external factuality evaluator. Conducting experiments with various base models, Mask-DPO outperforms vanilla-DPO and a existing factuality alignment method (FactTune) in both in-domain and out-of-domain datasets. Furthermore, it explores the generalization property of the method, which hypothesizes two corollaries to prove the characteristics of factuality alignment achieved by MASK-DPO.

**Strengths:**

**Reasonable Analysis of Factuality Alignment Through Data Scaling**

The study provides a well-founded hypothesis on the effects of factuality alignment by comparing outcomes based on the number of topics and questions. The subsequent proof-of-concept experiments offer a valid approach to support this hypothesis.

**Novelty of provided Methods"

The authors provide a novel mask-DPO method which hinders to learn unfactual text fragments lied in each sentence.

**Weaknesses:**

**Potential for Errors Due to Sentence-Level Annotation**

The paper ranks responses through sentence-level evaluation, but this method may not always accurately reflect factuality. Depending on the type of sentence or factual argument, sentence-level evaluation may misalign with actual factual accuracy. For instance, consider two responses conveying the same information:
        - Response 1: Consists of 10 short sentences.
        - Response 2: Consists of 3 long sentences.
If hallucination occurs in only one part of each response, the annotation scores could vary significantly (e.g., 9/10 vs. 1/3). Factors such as sentence segmentation and fact placement can impact response ratings, introducing potential errors in sentence-level evaluation.

**concerns about overall quality of the responses**

When it comes to response coherence, it raises concerns that the quality of the response may be compromised when certain sentences are masked. While masking can improve factual accuracy by filtering out incorrect signals from sampled responses, it risks disrupting the overall coherence and completeness of the response, as preference learning is based on responses where specific sentences have been removed partway through. Therefore, it may potentially leads to unexpected or lower-quality outputs even if they are factually correct. I understand that the primary focus of the paper is on the factuality of model responses, yet factuality itself aims to enhance overall response quality in LLMs.  Given this, I wonder if there is any evaluation or consideration of how the sentence removal approach in Mask-DPO's preference data construction might negatively impact response quality.

**insufficient details**

1) Ranking and Pairwise selection process
2) Prompts
3) Section 4.3: Proof of concept
    - embedding details
    - clustering topic vectors
    - choosing the far group and the near group

**Questions:**

[missing references] Long-form factuality generation in related work

1) Long-form Factuality in Large Language Models (Wei et al., 2024)
2) OLAPH: Improving Factuality in Biomedical Long-form Question Answering (Jeong et al., 2024)

[Sampling strategy] In line 344, it is quite hard to understand the first sentence that explains about the non-policy model effectiveness.

---

> ### Author Response · Authors · 2024-11-21
> **Responses to Official Review by Reviewer upqN: Part 1**
>
> Thank you for your constructive comment. The following are our responses to each individual comment (highlighted in italics).
>
> ### **Response to Weakness1 about potential errors due to sentence-level annotation:**
>
> > *The paper ranks responses through sentence-level evaluation, but this method may not always accurately reflect factuality. Depending on the type of sentence or factual argument, sentence-level evaluation may misalign with actual factual accuracy. For instance, consider two responses conveying the same information: - Response 1: Consists of 10 short sentences. - Response 2: Consists of 3 long sentences. If hallucination occurs in only one part of each response, the annotation scores could vary significantly (e.g., 9/10 vs. 1/3). Factors such as sentence segmentation and fact placement can impact response ratings, introducing potential errors in sentence-level evaluation.*
>
> We acknowledge the limitation of sentence-level hallucination annotations, but we do not think that this actually affects the effectiveness of our approach. Thanks to the mask mechanism, Mask-DPO allows the model to learn the correct signals even if the factuality scores of the preferred sample are lower than the non-preferred sample, since the wrong content in the preferred sample and the correct content in the non-preferred sample are masked out.
>
> Moreover, the form of Mask-DPO itself is general and does not assume that only sentence-level annotations can be used. The results in the paper show that using finer-grained annotations, such as sentence-level annotations, can lead to better results than using response-level annotations.  If an external hallucination annotator can provide more detailed annotations, such as at the level of clauses, noun phrases, verb phrases, etc., we believe the results will be better. However, there is currently a lack of hallucination annotators that can provide more fine-grained information. So we believe that the bottleneck actually lies in the external hallucination annotator instead of our approach.
>
> We have added a corresponding discussion in the Limitation part of our revised manuscript (Appendix I, line 905-911).
>
> ### **Response to Weakness2 about the overall quality of the responses:**
>
> > *When it comes to response coherence, it raises concerns that the quality of the response may be compromised when certain sentences are masked. While masking can improve factual accuracy by filtering out incorrect signals from sampled responses, it risks disrupting the overall coherence and completeness of the response, as preference learning is based on responses where specific sentences have been removed partway through. Therefore, it may potentially leads to unexpected or lower-quality outputs even if they are factually correct. I understand that the primary focus of the paper is on the factuality of model responses, yet factuality itself aims to enhance overall response quality in LLMs. Given this, I wonder if there is any evaluation or consideration of how the sentence removal approach in Mask-DPO's preference data construction might negatively impact response quality.*
>
> We wish to clarify that our approach does not affect the quality of the responses, such as overall consistency and completeness.  To avoid false signals, we do mask certain sentences and exclude them from the loss calculation. However, for sentences that require loss and gradient computation, the generation conditions remain the same regardless of whether the preceding sentences are masked or not (cf. Eq. 5). Therefore, we believe that our approach does not destroy the integrity of the response.
>
> To further demonstrate this, we add a text quality evaluation, using the GPT-2 to compute the average perplexity of text generated by the base model (Llama3.1-8B-Instruct) and the model after Mask-DPO training.
>
> Evaluated on the ANAH test set, the average PPL of the base model is **44.20**, while after Mask-DPO training the average PPL is **34.91**, suggesting that the training does not affect the quality of the generated text, and even makes it better.
>
> We have added a corresponding discussion and evaluation results in our revised manuscript (Appendix G, line 879-887).

---

> ### Author Response · Authors · 2024-11-21
> **Responses to Official Review by Reviewer upqN: Part 2**
>
> ### **Response to Weakness3 about the details:**
>
> > *Ranking and Pairwise selection process; Prompts; embedding details; clustering topic vectors; choosing the far group and the near group*
>
> **Details about** **ranking and pairwise selection process.**
>
> For a single question, we sample multiple responses from the model. For a response, we first split it into several sentences, and then annotate them sentence by sentence using the hallucination annotator. We choose ANAH-v2 [1] as the annotator, which provides four types of labels for each sentence, i.e., ‘No Facts’, ‘No Hallucination’, ‘Contradictory’ and ‘Unverifiable’. Following the same setup as in [1], we treat the first two types as hallucination-free and the last two as hallucination. We take the proportion of sentences in a response that is hallucination-free as the factuality score of that response. Based on the factuality score, we rank between multiple responses and select the highest-scoring and lowest-scoring responses to form preference pairs.
>
> **Details about prompt.**
>
> We do not use any special prompts other than the hallucination annotations. Both in the construction of the preference data and in the evaluation process, we directly use the question in the dataset as a prompt. For hallucination annotations, as we have expressed in Appendix B, we use the prompts provided by [1], in particular the Figures A1-A3 of [1].
>
> **Details about embedding, clustering and choosing the far group and the near group.**
>
> For each topic, we use the output of the last hidden layer of the model as its embedding vector. The embedding vectors of the topics in the test set are treated as multiple cluster centroids. For each topic in the training set, we calculate the Euclidean distance from its embedding vector to each cluster centroid. Each training set topic is then assigned to the closest cluster centroid, forming a series of clusters.
>
> Within each cluster, the training set topics are further divided into two groups—far and near—based on their distances to the cluster centroid. Topics closer to the centroid are grouped into the near group, while those farther away are grouped into the far group.
>
> Finally, the far groups from all clusters are merged into a single large far group, and similarly, the near groups are merged into a single near group. Based on the group to which a topic belongs, preferred pairs are reorganized into two separate training datasets. These datasets are then trained separately.
>
> We have added these details in our revised manuscript (Appendix A and B, line 710-740).
>
> ### **Response to Question1 about the missing references:**
>
> > *Long-form factuality generation in related work*
> >
> > 1. *Long-form Factuality in Large Language Models (Wei et al., 2024)*
> > 2. *OLAPH: Improving Factuality in Biomedical Long-form Question Answering (Jeong et al., 2024)*
>
> Thanks for your suggestion! LongFact [2] and OLAPH [3] both provide long-form question datasets, with OLAPH being more focused on the biomedical domain and LongFact encompassing multiple domains. And, OLAPH also suggests a simple and efficient framework to reduce hallucinations and include crucial medical claims. We have added these relevant papers to our related work in the revised version.
>
> We have added a corresponding discussion in the related work part of our revised manuscript (Section5, line 503-505).
>
> ### **Response to Question2 about the unclear presentation:**
>
> > *In line 344, it is quite hard to understand the first sentence that explains about the non-policy model effectiveness.*
>
> The non-policy implies that the model used to generate the preference data is different from the model that is ultimately trained. For example, for the first row in Table3, the preference pairs constructed using responses generated by the non-policy model (nternLM2-7B-Chat-SFT) are used to train the policy model (Llama3.1-8B-Instruct) for factuality alignment. We observe that this setup is not efficient enough, and this observation is also mentioned in other DPO-related paper [4]. Therefore, we suggest that the preference data used to perform alignment training should be sampled from the policy model.
>
> [1] Gu Y, Ji Z, Zhang W, et al. Anah-v2: Scaling analytical hallucination annotation of large language models[J]. arXiv preprint arXiv:2407.04693, 2024.
>
> [2] Wei J, Yang C, Song X, et al. Long-form factuality in large language models[J]. arXiv preprint arXiv:2403.18802, 2024.
>
> [3] Jeong M, Hwang H, Yoon C, et al. OLAPH: Improving Factuality in Biomedical Long-form Question Answering[J]. arXiv preprint arXiv:2405.12701, 2024.
>
> [4] Lin S C, Gao L, Oguz B, et al. Flame: Factuality-aware alignment for large language models[J]. arXiv preprint arXiv:2405.01525, 2024.

---

> ### Comment · Reviewer_upqN · 2024-11-23
>
> Thank you for your detailed explanation. Here's are my following question and fixing a minor detail.
>
> >  We choose ANAH-v2 [1] as the annotator, which provides four types of labels for each sentence, i.e., ‘No Facts’, ‘No Hallucination’, ‘Contradictory’ and ‘Unverifiable’. Following the same setup as in [1], we treat the first two types as hallucination-free and the last two as hallucination.
>
> In lines 710–717, it is mentioned that the hallucination annotator ANAH-v2 uses four label types. Among these, **No Facts** and **No Hallucination** are assumed to be hallucination-free, while **Contradictory** and **Unverifiable** are considered to contain hallucinations. In this context, I believe that regardless of the presence of hallucinations, there may be differences in the types of hallucinations between the **Contradictory** and **Unverifiable** labels. Have you conducted any additional analysis regarding this?
>
> > Appendix G, line 879-887
>
> I guess the title of the Appendix G has a typo (disccusion to discussion)

---

> > ### Author Response · Authors · 2024-11-23
> > **Responses to Official Comment by Reviewer upqN**
> >
> > Thank you for your response and comment. The following are our responses to each comment.
> >
> > ### Response to the question about details:
> >
> > > *In lines 710–717, it is mentioned that the hallucination annotator ANAH-v2 uses four label types. Among these,* ***No Facts*** *and* ***No Hallucination*** *are assumed to be hallucination-free, while* ***Contradictory*** *and* ***Unverifiable*** *are considered to contain hallucinations. In this context, I believe that regardless of the presence of hallucinations, there may be differences in the types of hallucinations between the* ***Contradictory*** *and* ***Unverifiable*** *labels. Have you conducted any additional analysis regarding this?*
> >
> > These two types of hallucinations do have subtle differences. According to the settings described in the ANAH-v2 [1], “Contradictory” refers to generated content that conflicts with the reference document, while “Unverifiable” means that no basis can be found in the reference document, though this definition does not rule out the possibility that the content may be correct. Consequently, our classification of “Unverifiable” as a type of hallucination, which is consistent with the definitions provided in the ANAH-v2 paper [1], may introduce some noise.
> >
> > Nevertheless, this can also be interpreted as learning a stricter preference: we aim for the model to generate content that is supported by mainstream reference documents. To evaluate the impact of this preference, we measured the percentage of “Unverifiable” sentences generated by the model on the test set before and after training. The results show a decrease from **39.27%** to **18.32%**, suggesting that our factuality alignment strategy improves the model’s ability to generate evidence-based content.
> >
> > While we acknowledge that using “Unverifiable” data is not optimal, we believe, as discussed in our previous discussion of sentence-level annotations, that this limitation stems from the external hallucination annotator rather than from Mask-DPO itself.
> >
> > ### Response to the question about typo:
> >
> > > *I guess the title of* *the* *Appendix G has a typo (disccusion to discussion)*
> >
> > Yes, there is a typo. We have made a correction. Thank you for the heads up.
> >
> > [1] Gu Y, Ji Z, Zhang W, et al. Anah-v2: Scaling analytical hallucination annotation of large language models[J]. arXiv preprint arXiv:2407.04693, 2024.

---

> ### Comment · Reviewer_upqN · 2024-11-25
>
> From the reviewer's perspective, it seems inevitable that MaskDPO heavily relies on the hallucination annotator (currently at the sentence level). Nevertheless, it is evident that MaskDPO effectively enhances factuality in terms of both extrinsic hallucination (i.e., contradictory) and intrinsic hallucination (i.e., unverifiable) while enhancing the generation ability (may be limited qualified in ANAH dataset). Taking these points into account, I will proceed to update the scores accordingly.

---

> > ### Author Response · Authors · 2024-11-25
> >
> > Thank you for your response and for increasing the rating. We are happy that our discussions on algorithm performance, implementation details and related works are convincing. We acknowledge that Mask-DPO, as a preference learning algorithm, inevitably relies on external reward models, which is also present in other xPO algorithms. We view Mask-DPO as a first step towards fine-grained factuality alignment and hope for more research on this topic in the future.

---

### Official Review · Reviewer_VhTU · 2024-11-04

**Soundness:** 3
**Presentation:** 3
**Contribution:** 3
**Rating:** 6
**Confidence:** 4

**Summary:**

This paper proposes a novel fine-grained factuality alignment method called Mask-DPO to mitigate hallucinations in large language models (LLMs). Mask-DPO incorporates sentence-level factuality information as a mask signal to guide the preference learning, avoiding the reward for incorrect sentences in preferred samples and the penalty for factual sentences in non-preferred samples. Experiments demonstrate that Mask-DPO significantly improves the factuality of LLM responses on both in-domain and out-of-domain datasets, even when the test questions and topics are unseen during training. The authors also study the generalization property of Mask-DPO and find that scaling the number of topics in the training data is more effective than scaling the number of questions. They hypothesize an internal knowledge graph learned by LLMs and provide proof-of-concept experiments to verify it.

**Strengths:**

1. Mask-DPO addresses a critical issue of hallucination in LLMs, which hinders their practical applications in various domains. The proposed method provides a more effective solution compared to existing response-level preference learning approaches by leveraging fine-grained sentence-level factuality signals.

2. The paper provides valuable insights into the generalization property of Mask-DPO by studying different training data scaling strategies. The finding that scaling the number of topics is more effective than scaling the number of questions sheds light on how to efficiently construct training data for factuality alignment.

3. The hypothesis of an internal knowledge graph learned by LLMs during pre-training is intriguing and opens up new research directions. The authors provide proof-of-concept experiments to support this hypothesis, such as ablating different topic sampling strategies and comparing the factuality of best-of-N responses before and after alignment.

**Weaknesses:**

1. While the paper compares Mask-DPO with DPO, it would be beneficial to include a more comprehensive comparison with other state-of-the-art factuality alignment methods. This would help to better position Mask-DPO in the context of existing approaches and highlight its unique advantages.

2. The hypothesis of the internal knowledge graph learned by LLMs is intriguing but lacks sufficient theoretical justification and empirical evidence. The authors could provide a more rigorous theoretical analysis of how the knowledge graph is formed during pre-training and how it relates to the factuality alignment process. Additionally, visualizing the graph structure or quantitatively measuring the affinity between nodes would further support the hypothesis.

3. The paper would benefit from a more thorough proofreading to fix minor typos and grammatical errors (e.g., "anachronistic sentences" in Figure 2 caption, "preferece" in line 169 of Section 2.1).

**Questions:**

Section 3.1: It would be helpful to provide more details on the specific subset of the Biography dataset used for out-of-domain evaluation, such as the criteria for selecting the questions and the diversity of the topics covered.

Section 3.2: The authors could consider adding a case study or qualitative analysis of the generated responses before and after applying Mask-DPO. This would give readers a more intuitive understanding of how Mask-DPO improves factuality at the sentence level.

---

> ### Author Response · Authors · 2024-11-21
> **Responses to Official Review by Reviewer VhTU: Part 1**
>
> Thank you for your constructive comment. The following are our responses to each individual comment (highlighted in italics).
>
> ### **Response to Weakness1 about comparison with other sota methods:**
>
> > *While the paper compares Mask-DPO with DPO, it would be beneficial to include a more comprehensive comparison with other state-of-the-art factuality alignment methods. This would help to better position Mask-DPO in the context of existing approaches and highlight its unique advantages.*
>
> We wish to clarify that we have the comparison with other state-of-the-art factuality alignment method, i.e., FactTune [1] (penultimate row of Table1), which was published in ICLR2024.
>
> Based on your suggestion, we supplement the comparison with another state-of-the-art factuality alignment method, i.e., Flame [2], which was published in Neurips2024. Specifically, we construct factuality preference data on our training set and perform SFT+DPO following the method described in their paper. We use Llama3.1-8B-Instruct as the base model and use ANAH-v2 for evaluation.
>
> The following table reports the comparison between Mask-DPO and the other two SOTA methods, where Mask-DPO gives better results.
>
> |          | # Correct | #Incorrect | % Score |
> | -------- | --------- | ---------- | ------- |
> | FactTune | 657       | 499        | 56.83   |
> | Flame    | 447       | 263        | 62.96   |
> | Mask-DPO | 547.6     | 161.4      | 77.53   |
>
> We want to emphasize that the main focus of all other factuality alignment methods is on the construction of the preference data, but the alignment methods themselves are all DPO. However, we are aware of the limitations of DPO for factuality tasks and make targeted modifications.
>
> We have added a corresponding discussion and evaluation results in our revised manuscript (Appendix C, line 782-798).
>
> ### **Response to Weakness2 about theoretical analysis of hypothesis:**
>
> > *The hypothesis of the internal* *knowledge graph* *learned by LLMs is intriguing but lacks sufficient theoretical justification and empirical evidence. The authors could provide a more rigorous theoretical analysis of how the knowledge graph is formed during pre-training and how it relates to the factuality alignment process. Additionally, visualizing the graph structure or quantitatively measuring the affinity between nodes would further support the hypothesis.*
>
> The idea-there is some structure in language modeling-has been explored in previous classic works (e.g., [4], [5], [6]), which argue that language models produce similar representations of words with high affinities. In other words, for an ideal language model, the relevance of the semantic space and the similarity of the semantic space represent spaces that are consistent.
>
> Building on this understanding, we propose the internal knowledge structure hypothesis. As discussed in Section 4.2, we hypothesize that the model's blurred recognition of similar topics leads to the mixed use of their respective subordinate content, resulting in phenomena akin to hallucination (cf. Eq 8). We conceptualize factual alignment as a process of adjusting the relationships between topics. During the alignment of a given topic, the affinity between the topic and its subordinate content increases, while the affinity with the subordinate content of similar topics decreases. This ultimately manifests as a reduction in hallucinated content.
>
> We randomly selected 200 data from the preference dataset used for training to perform a quantitative analysis of affinities. Specifically, for each preference data point, we extracted the entities from both the chosen response and the rejected response, embedding these entities using word embeddings. We then calculated the average distance between the word vectors of these entities and the vector of the topic to which the preference data belongs. This evaluation was performed on the model before and after training separately.
>
> The results indicate that, after training, the average distance between the topic vector and the entities in the chosen response decreased by **1.97**, while the average distance to the entities in the rejected response increased by **4.09**. These findings support the reliability of our affinity-based explanation of the internal knowledge structure.
>
> We have added a corresponding discussion and evaluation results in our revised manuscript (Appendix F, line 841-863).

---

> ### Author Response · Authors · 2024-11-21
> **Responses to Official Review by Reviewer VhTU: Part 2**
>
> ### **Response to Weakness3 about typos:**
>
> > *The paper would benefit from a more thorough proofreading to fix minor typos and grammatical errors (e.g., "anachronistic sentences" in Figure 2 caption, "preferece" in line 169 of Section 2.1).*
>
> Thanks for your suggestion! We have fixed the typos you mentioned in our revised manuscript.
>
> ### **Response to Question1 about details on Biography:**
>
> > *It would be helpful to provide more details on the specific subset of the Biography dataset used for out-of-domain evaluation, such as the criteria for selecting the questions and the diversity of the topics covered.*
>
> We use the Biography dataset from [3], which includes 683 Wikipedia person entries. The task is to generate biographies based on queries in the format: “Question: Tell me a bio of XXX.” Following the split method in [2], we selected 183 entries for testing. The selection was randomized under the condition that no overlap exists with the ANAH dataset used for training, ensuring an OOD (Out-Of-Distribution) testing environment.
>
> ### **Response to Question2 about the** **case study:**
>
> > *The authors could consider adding a case study or qualitative analysis of the generated responses before and after applying Mask-DPO. This would give readers a more intuitive understanding of how Mask-DPO improves factuality at the sentence level.*
>
> Thanks for your suggestion! We have added a case study in our revised manuscript (Appendix H, line 888-893).
>
>
>
> [1] Tian K, Mitchell E, Yao H, et al. Fine-tuning language models for factuality[J]. arXiv preprint arXiv:2311.08401, 2023.
>
> [2] Lin S C, Gao L, Oguz B, et al. Flame: Factuality-aware alignment for large language models[J]. arXiv preprint arXiv:2405.01525, 2024.
>
> [3] Min S, Krishna K, Lyu X, et al. Factscore: Fine-grained atomic evaluation of factual precision in long form text generation[J]. arXiv preprint arXiv:2305.14251, 2023.
>
> [4] Mikolov T, Sutskever I, Chen K, et al. Distributed representations of words and phrases and their compositionality[J]. Advances in neural information processing systems, 2013, 26.
>
> [5] Mikolov T. Efficient estimation of word representations in vector space[J]. arXiv preprint arXiv:1301.3781, 2013, 3781.
>
> [6] Bengio Y, Ducharme R, Vincent P. A neural probabilistic language model[J]. Advances in neural information processing systems, 2000, 13.

---

### Official Review · Reviewer_bLBp · 2024-11-07

**Soundness:** 3
**Presentation:** 3
**Contribution:** 3
**Rating:** 6
**Confidence:** 4

**Summary:**

In this paper, the authors introduce an alignment method to improve factuality in LM responses.
The method, based on DPO, leverages factuality annotations at the sentence level to avoid optimizing non-factual sentences in chosen responses and to avoid penalizing factually correct sentences in rejected responses during preference optimization.
Experiments in and out of domain demonstrate the effectiveness of the strategy on improving response factuality.
Most intriguing, the authors provide a hypothesis, accompanied with preliminary experiments, about how factuality alignment influence topic representation and the generalization capabilities of an LM.

**Strengths:**

S1. The proposed method is well presented and supported mathematically.

**Weaknesses:**

W1. Even though the experimental setup includes a wide range of model families, it does not include transparent SFT+PO (instruction tuning + preference optimization) training setups, training PO directly over instruct models. Previous work in PO encourages starting from a base model (not instruction-tuned), perform SFT over chosen responses, and then perform PO. Such setup provides high transparency on the training procedure of a final chat / PO model. For reference, please refer to the DPO paper.

W2. There is a lack of comparison against relevant, recent PO strategies. Similar work that implement some form of masking or granularity below response are: TDPO, DPOP, SePO, to name a few.

W3. Whilst the paper presents extensive results on factuality metrics, a more comprehensive evaluation on standard benchmarks is needed. In this way, we could appreciate the effect of factuality alignment over other LLM capabilities, e.g. mathematical reasoning or harmful generation.

[1] https://arxiv.org/abs/2404.11999
[2] https://arxiv.org/pdf/2402.13228
[3] https://arxiv.org/pdf/2408.13518

**Questions:**

## Appendix A
-	Regarding the inference setup, temperature=0.8, was hyper-parameter tuning performed? Ideally, it would be helpful to show how performance varies across temperature values (e.g. 0, 0.25, 0.5, 0.75, 1.0), as it is done in PO literature.

## Granularity of Annotation
In the paper, annotation is gathered at the sentence level. Would it be possible to apply this same annotation procedure to more fine-grained syntactic units? For instance, clauses, noun phrases, verb phrases. Do you have an intuition as to how impactful such a setup would be?

## Reporting results
-	Was a statistical significance test performed on the reported metric results?
-	I am not familiar with ANAH and FactScore metrics, would it be safe to assume they follow a gaussian distribution? If not, it is important to consider that, in cases where the metric score distribution cannot be considered gaussian, it is better to report results over bootstrap resampling of the metric scores. Please refer to EleutherAI’s Evaluation Harness (lm-eval), a popular evaluation library for LLMs, for details.

[4] https://github.com/EleutherAI/lm-evaluation-harness

---

> ### Author Response · Authors · 2024-11-21
> **Responses to Official Review by Reviewer bLBp: Part 1**
>
> Thank you for your constructive comment. The following are our responses to each individual comment (highlighted in italics).
>
> ### **Response to Weakness1 about transparent SFT+PO:**
>
> > *Previous work in PO encourages starting from a base model (not instruction-tuned), perform SFT over chosen responses, and then perform PO. Such setup provides high transparency on the training procedure of a final chat / PO model.*
>
> We wish to emphasize that omitting SFT does not affect our conclusions. Since Mask-DPO is a general framework for fine-grained preference learning, it does not have any specific assumption or reliance on the SFT process. We also note that other related works on factuality alignment, such as [2] and [3], do not incorporate SFT+PO.
>
> We do agree with your suggestion that including such a setup could provide greater transparency on the training procedure. Thus, we add an experiment on SFT+Mask-DPO based on the setup described in the DPO paper [4]. Specifically, we apply SFT to the base model (not instruction-tuned), Llama3.1-8B, using the chosen responses from the existing preference data. Subsequently, we perform Mask-DPO on the supervised model.
>
> The table below presents the results of SFT+Mask-DPO and DPO under ANAH-v2 evaluation. The results show that executing SFT+Mask-DPO from the base model has similar results as executing Mask-DPO directly from the struct model. This also shows again that our fine-grained preference learning approach does not have any specific assumption or reliance on the SFT process.
>
> |                                  | # Correct | #Incorrect | % Score |
> | -------------------------------- | --------- | ---------- | ------- |
> | Llama-3.1-8B+ SFT + Mask-DPO     | 631       | 192        | 76.67   |
> | Llama-3.1-8B-Instruct + Mask-DPO | 547.6     | 161.4      | 77.53   |
>
> We have added a corresponding discussion and evaluation results in our revised manuscript (Appendix C, line 746-753).
>
> ### **Response to Weakness2 about comparison againt other PO Strategies:**
>
> > *There is a lack of comparison against relevant, recent PO strategies. Similar works that implement some form of masking or granularity below response are: TDPO, DPOP, SePO, to name a few.*
>
> In line with your suggestion, we conduct additional comparative experiments on TDPO [5], DPOP [6], and SePO [7]. The specific configurations are as follows:
>
> 1. TDPO: We use the default settings from their implementation, specifically the TDPO2 method with the alpha parameter set to 0.5.
> 2. DPOP: The lambda parameter is set to 50, consistent with the original paper.
> 3. SePO: We utilize the Oracle Model provided in their work to estimate the token-level reward function.
>
> For all experiments, we use Llama3.1-8B-Instruct as the base model and ANAH-v2 for evaluation.
>
> The comparison results are shown in the table below. The performance of TDPO and DPOP is similar to DPO, while SePO performs significantly worse than DPO. This discrepancy may be due to the Oracle Model used in SePO not being trained on our tasks, which limits its ability to accurately estimate the token-level reward.
>
> |          | # Correct | #Incorrect | % Score |
> | -------- | --------- | ---------- | ------- |
> | DPO      | 446.4     | 206        | 68.44   |
> | DPOP     | 593       | 291        | 67.08   |
> | TDPO     | 703.6     | 284        | 70.95   |
> | SePO     | 254       | 611.4      | 58.45   |
> | Mask-DPO | 547.6     | 161.4      | 77.53   |
>
> We have added a corresponding discussion and evaluation results in our revised manuscript (Appendix C, line 754-781).
>
> ### **Response to Weakness3 about evaluation on standard benchmarks:**
>
> > *We could appreciate the effect of factuality alignment over other LLM capabilities, e.g. mathematical reasoning or harmful generation.*
>
> The main focus of the paper is on factuality alignment, and we use only factuality preference data for training. As a result, we cannot guarantee strong performance on other types of benchmarks.
>
> We supplement the evaluation with relevant benchmarks for math and code. Specifically, we test the model on the GSM8K, MATH, GPQA, Human Eval, and MBPP datasets using the OpenCompass evaluation framework. The base model is Llama3.1-8B-Instruct.
>
> The table below presents the results of the model on these benchmarks before and after training. Mask-DPO shows a slight decrease in performance on one math benchmark (MATH) and two code benchmarks (Human Eval and MBPP), while demonstrating improvements on two math benchmarks (GSM8K and GPQA). These results suggest that factuality alignment methods may have marginal effects on the other capabilities of the model.
>
> |          | GSM8K | MATH  | GPQA  | Human Eval | MBPP  |
> | -------- | ----- | ----- | ----- | ---------- | ----- |
> | Baseline | 84.91 | 52.72 | 26.77 | 70.73      | 71.21 |
> | Mask-DPO | 86.20 | 47.68 | 31.82 | 68.29      | 70.04 |
>
> We have added a corresponding discussion and evaluation results in our revised manuscript (Appendix D, line 799-815).

---

> ### Author Response · Authors · 2024-11-21
> **Responses to Official Review by Reviewer bLBp: Part 2**
>
> ### **Response to Question1 about inference hyper-parameter:**
>
> > *Regarding the inference setup, temperature=0.8, was hyper-parameter tuning performed? Ideally, it would be helpful to show how performance varies across temperature values (e.g. 0, 0.25, 0.5, 0.75, 1.0), as it is done in PO literature.*
>
> We wish to clarify that we have not deliberately adjusted the temperature parameter. Following the configuration in [1], we set it to 0.8 to sample more diverse data.
>
> We conduct a series of experiments to investigate the effect of this parameter on performance. Specifically, we test four temperature settings: 0.25, 0.5, 0.75, and 1.0. We exclude 0 because sampling at this value produced insufficiently diverse data for our task, making it challenging to construct preference data. The other experimental conditions are consistent with those described in the paper. We use Llama3.1-8B-Instruct as the base model and evaluate performance using ANAH-v2.
>
> The comparison results are shown in the table below. Besides the default setting in our paper (0.8), the best performance is observed at temperature = 0.75, while performance at temperature = 0.25 is lower. Interestingly, this trend aligns with findings in the DPO paper [4], despite differences in task objectives. This suggests that the optimal sampling temperature for our task is around 0.8.
>
> | Temperature           | #Correct | #Incorrect | %Score |
> | --------------------- | -------- | ---------- | ------ |
> | 0.25                  | 485      | 259.2      | 65.30  |
> | 0.5                   | 539      | 244.8      | 67.96  |
> | 0.75                  | 733      | 219        | 76.02  |
> | 1.0                   | 732      | 240.6      | 74.24  |
> | 0.8 (default setting) | 547.6    | 161.4      | 77.53  |
>
> We have added a corresponding discussion and evaluation results in our revised manuscript (Appendix E, line 817-841).
>
> ### **Response to Question2 about the** **granularity of annotation:**
>
> > *In the paper, annotation is gathered at the sentence level. Would it be possible to apply this same annotation procedure to more fine-grained syntactic units? For instance, clauses, noun phrases, verb phrases. Do you have an intuition as to how impactful such a setup would be?*
>
> Yes, we expect Mask-DPO can be applied to these finer-grained syntactic units, because the form of Mask-DPO itself is general and does not assume that only sentence-level annotations can be used.
>
> The results in the paper show that using finer-grained annotations, such as sentence-level annotations, can lead to better results than using response-level annotations.  If an external hallucination annotator can provide more detailed annotations, such as at the level of clauses, noun phrases, verb phrases, etc., we believe the results will be better. However, there is currently a lack of hallucination annotators that can provide more fine-grained information. So we believe that the bottleneck actually lies in the external hallucination annotator instead of our approach.
>
> We have added a corresponding discussion in the Limitation part of our revised manuscript (Appendix I, line 905-911).
>
> ### **Response to Question3 about reporting results:**
>
> > *Was a statistical significance test performed on the reported metric results? I am not familiar with ANAH and FactScore metrics, would it be safe to assume they follow a gaussian distribution? If not, it is important to consider that, in cases where the metric score distribution cannot be considered gaussian, it is better to report results over bootstrap resampling of the metric scores.*
>
> We consider our results to be statistically significant because Table 2-5 reports the mean value after five replications. In the process of experimentation, we found that the results of repeated experiments are stable, so in the resource considerations for the baseline part of Table1 did not repeat the experiment.
>
> To further check whether the metrics follow a Gaussian distribution, we additionally report more statistical information for some key experiments using the Shapiro-Wilk method. Below is Table 2 of our paper, which contains the newly added statistics against the ANAH-v2 and Factcore metrics, i.e., the P-values (third and fifth columns). Under both ANAH-v2 and FactScore metrics, the P-values are significantly greater than 0.05, indicating that they follow a Gaussian distribution.
>
> |          | ANAH-v2 Score | ANAH-v2 P | FactScore Score | FactScore P |
> | -------- | ------------- | --------- | --------------- | ----------- |
> | DPO      | 68.44         | 0.493     | 23.43           | 0.355       |
> | Mask-DPO | 77.53         | 0.703     | 25.56           | 0.511       |

---

> ### Author Response · Authors · 2024-11-21
> **Responses to Official Review by Reviewer bLBp: Part 3**
>
> [1] Gu Y, Ji Z, Zhang W, et al. Anah-v2: Scaling analytical hallucination annotation of large language models[J]. arXiv preprint arXiv:2407.04693, 2024.
>
> [2] Tian K, Mitchell E, Yao H, et al. Fine-tuning language models for factuality[J]. arXiv preprint arXiv:2311.08401, 2023.
>
> [3] Zhang X, Peng B, Tian Y, et al. Self-alignment for factuality: Mitigating hallucinations in llms via self-evaluation[J]. arXiv preprint arXiv:2402.09267, 2024.
>
> [4] Rafailov R, Sharma A, Mitchell E, et al. Direct preference optimization: Your language model is secretly a reward model[J]. Advances in Neural Information Processing Systems, 2024, 36.
>
> [5] Zeng Y, Liu G, Ma W, et al. Token-level Direct Preference Optimization[J]. arXiv preprint arXiv:2404.11999, 2024.
>
> [6] Pal A, Karkhanis D, Dooley S, et al. Smaug: Fixing failure modes of preference optimisation with dpo-positive[J]. arXiv preprint arXiv:2402.13228, 2024.
>
> [7] Yang K, Liu Z, Xie Q, et al. Selective Preference Optimization via Token-Level Reward Function Estimation[J]. arXiv preprint arXiv:2408.13518, 2024.

---

> ### Comment · Reviewer_bLBp · 2024-11-25
>
> Thanks for addressing my comments.
> Please find a follow-up to the responses.
>
> > ### Response to Weakness1 about transparent SFT+PO:
>
> I agree that the rankings in model performance should not vary dramatically when using vs not using SFT.
> Regarding the results of SFT+PO vs PO, it seems that the ANAH-v2 scores slightly degrade when using SFT prior to PO.
> Do you have any insight on why this might be happening?
>
> > ### Response to Weakness2 about comparison against other PO Strategies:
>
> Thanks, these results paint a more complete picture.
>
> > ### Response to Weakness3 about evaluation on standard benchmarks:
>
> It is indeed interesting to see an improvement on GSM8K and GPQA, arguably two tasks in which a form of CoT is expected to be generated before arriving to the final answer. In this case, Mask-DPO might be encouraging more faithful reasoning steps.
>
> > ### Response to Question1 about inference hyper-parameter:
>
> Thank you for this analysis, it shows the robustness of the method to the sampling temperature. It would be interesting to see (in a future version perhaps) how the performance of Mask-DPO fares against DPO across temperatures.
>
> > ### Response to Question3 about reporting results:
> Thanks for the clarification experiment. This certainly makes the results more trustable.

---

> > ### Author Response · Authors · 2024-11-25
> >
> > Thank you for your response and for increasing the rating. We are happy that our discussions are convincing and address your concerns.
> >
> > Regarding slight performance degradation when using SFT+PO, we think that there could be several reasons: 1) Despite using the chosen response, it still includes hallucination content. 2) We observed that using only chosen responses for SFT, the trained model is less diverse in responses, which may affect preference data construction and preference learning.

---

### Author Response · Authors · 2024-11-24

We sincerely appreciate the valuable feedback from the reviewers!

We are honored that our work can be reviewed as:

- This paper addresses a critical issue of hallucination in LLMs (R-VhTU), significantly improving the results on the evaluated benchmarks (R-AbTR).
- The proposed method is novel (R-upqN), straightforward (R-GYLY), well presented and supported mathematically (R-bLBp).
- The paper provides valuable insights into the generalization property of Mask-DPO by studying different training data scaling strategies (R-VhTU).
- The study provides a well-founded hypothesis on the effects of factuality alignment and conducts experiments to validate these assumptions (R-upqN & R-GYLY & R-AbtR), which opens up new research directions (R-VhTU).

For each question from all reviewers, we have provided a specific response in the relevant section below and added corresponding discussion and evaluation results in our revised manuscript.

---

### Meta-Review · Area_Chair_fyHs · 2024-12-19

**Metareview:**

This paper introduces Mask-DPO, a novel fine-grained factuality alignment method that uses sentence-level factuality signals to improve preference learning and reduce hallucinations in LLMs. Mask-DPO outperforms state-of-the-art methods (e.g., DPO, FactTune, Flame) on multiple benchmarks and offers insights into scaling training data to enhance generalization.

Strengths:
- The authors provides a simple yet effective framework for fine-grained preference learning.
- This paper tackles the important challenge of hallucination in LLMs (R-VhTU) and achieves notable improvements on the evaluated benchmarks (R-AbTR).
- The method introduced is innovative (R-upqN), simple (R-GYLY), and both clearly presented and mathematically grounded (R-bLBp).
- The study presents a well-supported hypothesis on the impact of factuality alignment, validated through experiments (R-upqN, R-GYLY, R-AbTR).

Weaknesses:
- The proposed approach relies on sentence-level annotations. Reviewers raised concerns about the limitations of current hallucination annotators. In addition, cases where hallucination occurs within a single sentence remain challenging, making a bottleneck in annotation granularity rather than the method itself.

**Additional Comments On Reviewer Discussion:**

Reviewers rated the paper positively (scores: 6–8) and acknowledged its impact and robustness. Concerns about annotation limitations, missing comparisons with state-of-the-art baselines, and evaluation beyond factuality were raised but were addressed satisfactorily during the rebuttal.

The main concerns from the reviewers initially were: 1) annotation limitations, 2) missing comparisons with state-of-the-art baselines, and 3) expand evaluation beyond factuality. For annotation limitations, the authors acknowledged this as a constraint, which was accepted by the reviewers. Regarding missing comparisons with methods like TDPO and Flame, the authors conducted additional experiments demonstrating Mask-DPO’s consistent performance advantages. For evaluation beyond factuality, the authors addressed concerns by including results on GSM8K, GPQA, and coherence metrics, which highlighted Mask-DPO’s competitive performance.

---

### Decision · Program_Chairs · 2025-01-22

Accept (Poster)